# Model Selection of Discrete Classifiers under Class and Cost Distribution Change: An Empirical Study

## Abstract

A variety of important machine learning applications require predictions on test data with different characteristics than the data on which a model was trained and validated. In particular, test data may have a different relative frequency of positives and negatives (i.e., class distribution) and/or different mislabeling costs of false positive and false negative errors (i.e., cost distribution) than the training data. Selecting models that have been built in conditions that are substantially different from the conditions under which they will be applied is more challenging than selecting models for identical conditions. Several approaches to this problem exist, but they have mostly been studied in theoretical contexts. This paper presents an empirical evaluation of approaches for model selection under class and cost distribution change, based on Receiver Operating Characteristic (ROC) analysis. The analysis compares the ROC Convex Hull (ROCCH) method with other candidate approaches for selecting discrete classifiers for several UCI Machine Learning Repository and simulated datasets. Surprisingly, the ROCCH method did not perform well in the experiments, despite being developed for this task. Instead, the results indicate that a reliable approach for selecting discrete classifiers on the ROC convex hull is to select the model that optimizes the cost metric of interest on the validation data (which has the same characteristics as the training data) but weighted by the class and/or cost distributions anticipated at test time.

## 1 Introduction

Model selection can be a challenging aspect of machine learning practice. Reliable approaches for evaluating and selecting models have been the subject of ongoing research (Wagstaff, 2012; Flach, 2019; 2023). In supervised learning, the training, validation, and test data are usually assumed to be drawn from the same distribution (James et al., 2013; Kuhn et al., 2013). In particular, and with a focus on binary classification tasks, one typically assumes that the relative frequencies of positive and negative examples (i.e., the *class distribution*) and the relative costs of false positive and false negative errors (i.e., the *cost distribution*) are the same across datasets. However, sometimes test conditions are different from the training and validation conditions, in which case the testing data represents a *changed environment*. Methods to select models for changed environments exist, but they have mostly been analyzed theoretically. This paper contributes an empirical study of model selection approaches for binary classifiers in changed test environments.

A variety of impactful machine learning applications may exhibit changed environments at test time. Consider species distribution modeling (Elith & Leathwick, 2009), in which a binary classifier distinguishes between sites where a species of interest is present vs. absent based on environmental features. Species distribution models are important tools for scientists and managers to guide conservation policies. In this problem, the testing class distribution might change if a species is declining over time (i.e., fewer positive examples expected in predictions to the future), and the testing cost distribution might change based on the type of management action under consideration. A second example arises when predicting whether a climate model simulation will crash or not, in advance of the actual failure point (Lucas et al., 2013). Such predictions can save time and computational resources by terminating execution and restarting with appropriate modifications. In this application, larger simulations that crash are more costly than smaller simulations that crash. If larger simulations are anticipated in the future, then the realized false positive and

false negative costs would change accordingly. As a third example, consider a set of classifiers that predict whether or not a person is susceptible to heart disease (Detrano et al., 1989; Ayatollahi et al., 2019). We might be interested in selecting among these models and applying to data from a hospital where patients have higher rates of heart diseases than in the training data; in this case hospitals with differing patient demographics present changed environments. Given a set of classifiers to choose from, the approaches we investigate allow model selection for these changed environments.

Our problem setting applies to discrete classifiers, for which there may or may not be uncertainty regarding the anticipated testing conditions. Characteristics of changed environments, such as misclassification costs, are only applicable to discrete labels predicted by classifiers (Elkan, 2001). Therefore, model selection for changed environments depends on evaluation of discrete labels, and hence, discrete classifiers. Of course, continuous-output classifiers can be thresholded to obtain discrete-output classifiers (Zhao, 2008), in which case the corresponding continuous classifier may also be of interest. Sometimes, even though the testing environment is different from the training environment, its characteristics are known in advance; other times, there may be a range of possible testing conditions. Provost & Fawcett (2001) define *precise environments* as circumstances when we are precisely aware of class and cost distributions of test data (e.g., the test data will have 40% positives), and *imprecise environments* as circumstances in which we are uncertain about class and cost distributions but have a range of possibilities (e.g., the test data will have 30%-50% positives). Precise and imprecise environments can both be changed environments if their class and cost distributions are different from training conditions, and the model selection approaches studied herein can apply to both.

Note that the fields of cost-sensitive learning and imbalanced classification methods are related to our problem of interest but also orthogonal to our aim. These areas study supervised classification for imbalanced class distributions and classifiers that account for asymmetric cost distributions (Ling & Sheng, 2008). However, the standard setting for these approaches still assumes that the training and testing conditions are the same. In contrast, we focus on changed environments. Our experiments below select among classifiers that are insensitive to class imbalance and asymmetric costs, but the model selection approaches we investigate also apply to classifiers trained using imbalance-sensitive and cost-sensitive learning frameworks.

The central framework for model selection for changed environments to date has been Receiver Operating Characteristic (ROC) analysis (Flach, 2004; Fawcett, 2006). In particular, the ROC Convex Hull (ROCCH) method, described in detail below, is motivated in part by this problem setting (Provost & Fawcett, 1997; 1998; 2001). This method aims to choose the best performing classifier for a given set of operating conditions, from among the set of classifiers that comprise the ROC convex hull. The ROCCH method has been theoretically shown to select classifiers on the convex hull that minimize expected cost, but it has yet to be compared empirically to other approaches. In this paper, we compare the ROCCH method to simpler approaches for selecting discrete classifiers on the ROC convex hull, for changed environments.

Rigorous assessment of model selection approaches' practicality is necessary for deployment to real world systems. No previous work has investigated model selection of discrete classifiers under class and cost distributions change in an empirical study. Our paper aims to address this research gap. In particular, we contribute an evaluation of model selection approaches for selecting discrete classifiers on the ROC convex hull under class and cost distribution change by experimenting on 15 datasets from UCI Machine Learning Repository (Dua & Graff, 2017) and several simulated datasets. We draw attention to concerns about the ROCCH method and its reliance on isometrics, as we find that the method does not select the actual minimum cost classifier in most empirical settings. Our findings indicate that if we evaluate classifiers using normalized cost, then the most reliable approach is to select the classifier on the ROC convex hull which optimizes normalized cost on the validation set, but with the probabilities of making mistakes on positives and negatives on validation set weighted by cost distributions of changed test environments. Our experimental results also reveal that when model selection approaches choose high cost classifiers, those classifiers are often far from the optimal classifier in ROC space.

## 2    Background, Problem Statement, and the ROCCH Method

First, we briefly review some basics of ROC analysis; Section A.1 provides a glossary of terms for less familiar readers. ROC space is described by false positive rate ($FPR$) on the $x$-axis and true positive rate ($TPR$)

on the $y$-axis. Continuous classifiers (i.e., score predicting classifier $\hat{Y} \in \mathbb{R}$) can be thresholded to obtain discrete classifiers (i.e., discrete label predicting classifier $\hat{Y} \in \{0,1\}$), with their performance described by corresponding confusion matrices; thus, continuous classifiers form curves in ROC space, (*ROC curves*), while discrete classifiers appear as points. In Figure 1, Step 1 illustrates three ROC curves, with points defining discrete classifiers resulting from different thresholds. ROC curves are usually computed from predictions on validation data, from the same distribution as the training data. Discrete classifiers from different continuous classifiers lie on the same point in ROC space if they yield identical confusion matrices on the validation data. The ROC convex hull represents a hybrid classifier which corresponds to an ROC curve constructed from the most "north-western" available discrete classifiers in ROC space. Hybrid classifiers are composed of discrete classifiers that can originate from different continuous classifiers. In Figure 1, Step 2 shows the ROC convex hull of the three classifiers from Step 1. The QuickHull algorithm (Barber et al., 1993) can be used to generate the ROC convex hull of $n$ classifiers in $O(n \log n)$ time. The ROC convex hull has been shown to only contain the lowest cost classifiers for given intervals Provost & Fawcett (2001). When the testing environment matches the training and validation conditions, one might select the hybrid classifier or one of the discrete classifiers it comprises, which span a range of trade-offs between $FPR$ and $TPR$.

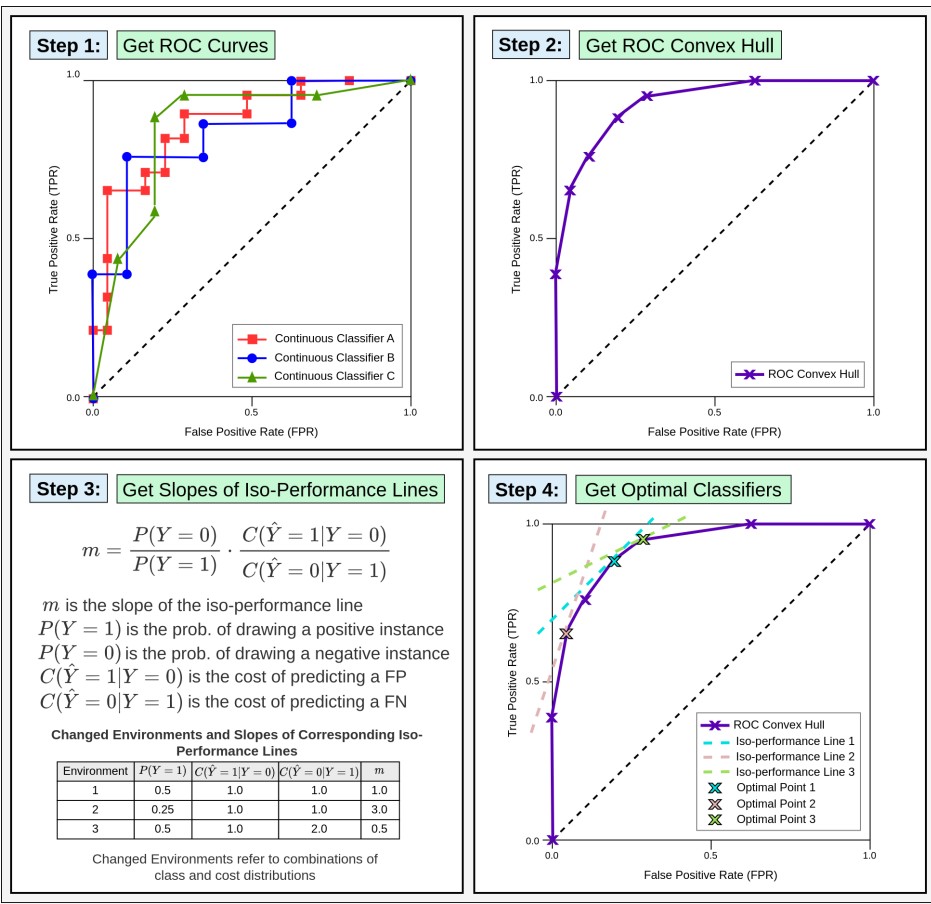

Figure 1: Illustration of some ROC concepts, and a conceptual example of applying the ROCCH method. Environments 1, 2 and 3 represent different changed environments characterized by their class and cost distributions. The method chooses a point on the hull based on pre-specified environmental conditions.

When the testing environment is anticipated to be different from the training and validation conditions, the available information on testing conditions should be incorporated into the model selection process. **More formally, this problem can be stated as follows: Given a set of discrete classifiers $S$ that are on the ROC convex hull, our task is to select the classifier(s) $s \in S$ that it optimizes some performance metric in a changed test environment, while assuming that $P_{train}(Y|X) =$**

$P_{test}(Y|X)$. We denote the discrete classifier that performs best on the test set as *Oracle*; this is the target classifier that we want to select. The problem of discrete classifier selection for changed environments was originally proposed by Provost & Fawcett (1997; 1998; 2001), and they introduced the ROCCH method for this task. Before describing their method, we must introduce the idea of isometrics in ROC space.

For some cost measures, the classifiers corresponding to different points in ROC space have equivalent performance. An *iso-performance line* is a line through ROC space such that all classifiers falling on that line have the same performance (Provost & Fawcett, 1997). To see this, consider the *expected cost* of a classifier

$$P(Y = 1) \cdot (1 - TPR) \cdot C(\hat{Y} = 0|Y = 1) + P(Y = 0) \cdot FPR \cdot C(\hat{Y} = 1|Y = 0), \tag{1}$$

with response variable $Y = 1$ for positives, $Y = 0$ for negatives, binary predictions $\hat{Y}$, and cost function $C$ (such that the false positive cost is $C(\hat{Y} = 1|Y = 0)$ and the false negative cost is $C(\hat{Y} = 0|Y = 1)$). Then two discrete classifiers represented by the ROC points $(FPR_1, TPR_1)$ and $(FPR_2, TPR_2)$ will have equal performance in terms of expected cost if

$$\frac{TPR_2 - TPR_1}{FPR_2 - FPR_1} = \frac{P(Y = 0) \cdot C(\hat{Y} = 1|Y = 0)}{P(Y = 1) \cdot C(\hat{Y} = 0|Y = 1)}.$$

With the objective of minimizing expected cost, the environmental conditions (i.e., class and cost distributions) can be encoded to a real value,

$$m = \frac{P(Y = 0) \cdot C(\hat{Y} = 1|Y = 0)}{P(Y = 1) \cdot C(\hat{Y} = 0|Y = 1)} = \left( \frac{P(Y = 0)}{P(Y = 1)} \right) \cdot \left( \frac{C(\hat{Y} = 1|Y = 0)}{C(\hat{Y} = 0|Y = 1)} \right).$$

This value represents the slope of an iso-performance line for expected cost (full derivation shown in Appendix Section A.2). For a given changed environment and a corresponding iso-performance line with slope $m$, the optimal discrete classifier for that environment will be the point on the ROC convex hull such that the iso-performance line with slope $m$ passes through that point and has highest $TPR$-intercept (Provost & Fawcett, 1997). An example showing the iso-performance line slopes for three changed environments is shown in Step 3 of Figure 1.

We now have all the pieces necessary to describe the *ROCCH method* (Provost & Fawcett, 1997; 1998; 2001), the four steps of which are illustrated in Figure 1):

1. Vary the classification threshold to generate (*FPR*, *TPR*) pairs for continuous classifiers;

2. Find the convex hull of all (*FPR*, *TPR*) points from all classifiers;

3. Find the slope of the iso-performance line corresponding to the performance metric of interest (typically, expected cost) for the anticipated class and/or cost distribution(s);

4. Select the discrete classifier at which the iso-performance line with the largest *TPR*-intercept intersects the convex hull.

Step 4 is carried out by iterating over each point on the convex hull and finding the point which leads to iso-performance line which largest possible $y$-axis ($TPR$) intercept. Provost & Fawcett (2001) showed that their ROCCH-hybrid minimizes expected cost for any given class and cost distribution because this hybrid is represented by the ROC convex hull comprised of optimal points for different $FPR$ and $TPR$ trade-offs. The ROCCH method relies on the assumption that the minimum expected cost classifier lies on iso-performance line corresponding to the computed slope.

The example above used expected cost as the performance metric, but *normalized cost* may also be of interest. Normalized cost measures empirical performance on the test set (Provost & Fawcett, 2001; Bettinger, 2003):

$$(1 - TPR) \cdot C(\hat{Y} = 0|Y = 1) + FPR \cdot C(\hat{Y} = 1|Y = 0). \tag{2}$$

This metric is 'normalized' since it does not directly include class distribution conditions, i.e., $P(Y = 1)$ and $P(Y = 0)$, in its calculation (as compared with expected cost in Equation 1 which does). Normalized cost

can be seen as a weighted sum of the probability of making mistakes on negatives $TNR$, and the probability of making mistakes on positives $TPR$, where the weights are the costs of those misclassifications. Drummond & Holte (2000) show how normalized cost (Equation 2) can be derived from expected cost.

## 3 Alternatives to the ROCCH Method

While the ROCCH method relies on isometrics to select a discrete classifier on the ROCCH built from validation data for a changed test environment, other approaches leverage the validation data by simply selecting the best performing model according to some metric. Since we still select from the models on the ROC convex hull, these other approaches follow Step 1 and 2 of the ROCCH method. Then, instead of using isometrics, we simply optimize some performance metric on the validation data. Algorithm 1 shows the general framework for metric-optimizing model selection approaches. This general approach is quite intuitive and commonplace in machine learning practice. Table 1 shows the changed environment conditions that are considered by different model selection methods.

---

**Algorithm 1** Metric-Optimizing Model Selection

---

**Input**: Metric $\Phi$, changed environment conditions $E$, discrete classifiers $S$, validation data $X_{val}, Y_{val}$
**Output**: Selected discrete classifier $s$

1: $n \leftarrow size(S)$                                   $\triangleright$ Number of discrete classifiers
2: $M \leftarrow \emptyset$                              $\triangleright$ For storing calculated metric values
3: **for** $i = 1, .., n$ **do**                      $\triangleright$ Loop over discrete classifiers
4:      $\hat{Y}_{val} \leftarrow S_i(X_{val})$              $\triangleright$ $S_i$ predicts binary labels on validation set
5:      $M_i \leftarrow \Phi(Y_{val}, \hat{Y}_{val}, E)$       $\triangleright$ $M_i$ stores metric value of classifier $S_i$ on $X_{val}, Y_{val}$
6: $s \leftarrow optimize(M, S)$       $\triangleright$ Select classifier (from $S$) which optimizes metric values in $M$

---

Table 1: Candidate approaches of model selection for changed environments.

| No. | Candidate Approach | Changed Environment Conditions | |
|---|---|---|---|
| | | Class Distribution | Cost Distribution |
| 1 | *ROCCH method* | ✓ | ✓ |
| 2 | *Norm-Cost-Min* | ✕ | ✓ |
| 3 | *Exp-Cost-Min* | ✓ | ✓ |
| 4 | *Accuracy-Max* | ✕ | ✕ |
| 5 | *F1-score-Max* | ✕ | ✕ |

The critical choice when applying Algorithm 1 is the performance metric $\Phi$; different metrics and their properties will lead to different behavior. In the experiments below, we investigated four variants of this algorithm, named for the metrics they optimize:

- *Norm-Cost-Min*: This model selection approach picks discrete classifier incurring lowest normalized cost (Equation 2) on validation set. This approach explicitly incorporates cost distribution information of changed environment into its calculation. Thus, $FPR$ and $TPR$ (and $FNR$) are calculated on the validation set, and weighted by cost distribution of changed test environment, i.e., $C(\hat{Y} = 1 | Y = 0)$ and $C(\hat{Y} = 0 | Y = 1)$ respectively, and then summed.

- *Exp-Cost-Min*: This model selection approach picks discrete classifier incurring lowest expected cost (Equation 1) on validation set. Though the expected cost has been viewed as theoretical while the normalized cost has been viewed as empirical (Provost & Fawcett, 2001; Bettinger, 2003), we still investigate this model selection approach based on optimizing expected cost. This method is aware of both class and cost distributions of changed environments.

- *Accuracy-Max*: This model selection approach picks discrete classifier that maximizes accuracy on validation set. This method has no prior knowledge about the class and/or cost distributions

of changed environments. Optimal discrete classifiers are picked solely based on performance on validation set.

- *F1-score-Max*: This model selection approach picks discrete classifier that maximizes F1-score on validation set. Similar to *Accuracy-Max*, this approach can not explicitly incorporate prior knowledge about class and cost distributions of changed test environments..

Note that different metrics $\Phi$ vary in how they incorporate changed environment conditions $E = \{P(Y = 1), P(Y = 0), C(\hat{Y} = 1|Y = 0), C(\hat{Y} = 0|Y = 1)\}$, as summarized in Table 1. For example, when $\Phi$ is *Accuracy-Max* or *F1-score*, the metric calculation ignores $E$. When $\Phi$ is *Normalized Cost*, the metric calculation uses the cost components of $E$ but not the class distribution, and when $\Phi$ is *Expected Cost*, all four components of $E$ are factored in.

## 4 Experimental Design

We studied the empirical performance of the ROCCH method and the alternatives of Section 3 by applying them to model selection problems on real and simulated datasets, using several modeling frameworks and evaluation approaches. In this section, we describe the datasets, how we manipulated class and cost distributions to generate changed test environments, and the details of our experiments. A schematic illustrating the overall methodology is in the Appendix (Figure 4). Our code will be made available upon acceptance of the paper via GitHub.

### 4.1 Datasets

We evaluated model selection approaches on simulated and real datasets. The simulated data generation process is described in the appendix (Section A.8). The 15 real datasets were from the UCI Machine Learning Repository (Table 2). Instances with missing features were removed, and categorical features were one-hot encoded. We split each dataset into (i) training data used for training continuous (score predicting) classifiers, (ii) validation data used for model selection, and (iii) test data which we manipulated to represent a changed environment (described further in Section 4.2). The proportions of instances falling into the training, validation, and testing sets varied in the range $\{0.2, 0.4, 0.6\}$ (e.g., a split could be 40% training data, 40% validation data, and 20% testing data as shown in Appendix Figure 5). After splitting, the training, validation, and test sets were scaled using the mean and standard deviation of the training set features, ensuring no information leakage. The process of splitting each dataset into three portions was repeated 30 times, and results were aggregated over all runs (discussed further in Section 4.3).

Table 2: UCI Machine Learning Repository datasets on which we evaluated model selection approaches.

| No. | Dataset | Instances | Features | Categorical Features | Class Balance |
|-----|---------|-----------|----------|----------------------|---------------|
| 1 | banknote-authentication | 1371 | 4 | 0 | 0.4449 |
| 2 | credit-approval | 652 | 15 | 9 | 0.4525 |
| 3 | german-credit | 999 | 24 | 0 | 0.6997 |
| 4 | australian-credit | 689 | 14 | 8 | 0.4456 |
| 5 | audit | 775 | 17 | 1 | 0.6271 |
| 6 | spambase | 4600 | 57 | 0 | 0.3939 |
| 7 | heart-disease | 296 | 13 | 0 | 0.4628 |
| 8 | heart-failure | 299 | 12 | 0 | 0.3211 |
| 9 | parkinsons | 195 | 23 | 0 | 0.7538 |
| 10 | habermans-survival | 305 | 3 | 0 | 0.7344 |
| 11 | mushroom | 5643 | 22 | 22 | 0.3819 |
| 12 | raisin | 900 | 7 | 0 | 0.5 |
| 13 | climate-model-crashes | 540 | 20 | 0 | 0.9148 |
| 14 | ionosphere | 350 | 34 | 0 | 0.64 |
| 15 | tic-tac-toe | 957 | 9 | 9 | 0.6531 |

### 4.2 Changed Environments

**Changing Class Distributions.** It is obvious that class distribution change in previously unseen real-world data is not trivially imitable. However, we replicated this phenomenon synthetically by manipulating the class distributions of changed test environments to our experiments' requirements. First, we stratified the partitioning of the data by $Y$ so that $P_{test}(Y = 1) \approx P_{train}(Y = 1)$. Then, depending on whether $\frac{P_{test}(Y=1)}{P_{train}(Y=1)}$, is above or below 1, either the positives needed to be oversampled and negatives undersampled or vice versa to produce a test set with the desired properties. This enabled us to tune the class distribution of changed environments, while ensuring that the total number of instances in the test set remains unchanged. We used two approaches to produce label shift: (i) *Random*, and (ii) *SMOTE-NearMiss*. Under *Random*, the oversampled instances were duplicates of randomly selected instances and undersampled instances were deleted. Under *SMOTE-NearMiss*, instances were oversampled with Synthetic Minority Oversampling TEchnique (SMOTE) (Chawla et al., 2002) and undersampled using NearMiss v1 (Mani & Zhang, 2003). SMOTE creates artificial instances, which introduce more variability in the feature space of changed test environments (compared to random oversampling which simply produces duplicates). Figure 6 in the Appendix illustrates our approach.

**Changing Cost Distributions.** Cost distributions of changed testing environments can be set during model selection, and require no additional processing of test data. In real-world settings where the positive class is of more interest, missed alarms (false negatives) are often considered worse than false alarms (false positives). Therefore, we keep $C(\hat{Y} = 1|Y = 0) = 1.0$ and vary $C(\hat{Y} = 0|Y = 1)$ in the range $\{1.0, 1.5, 3.0\}$. It is interesting to note that the slope of the iso-performance line, and hence the optimal classifier on the convex hull selected by the *ROCCH method*, depends on the cost ratio; therefore, for any $(C(\hat{Y} = 1|Y = 0), C(\hat{Y} = 0|Y = 1))$ pair, if we scale both misclassification costs using the same constant, we will always end up with the same iso-performance line and optimal discrete classifier.

### 4.3 Model Fitting

To generate a set of models to be selected from, we fit continuous classifiers from the `scikit-learn` package (Pedregosa et al., 2011). The modeling frameworks were logistic regression, $k$-nearest neighbor, and random forest, with default parameters. Random seed values were used to make results reproducible. All three continuous classifiers can predict posterior probabilities i.e., $\hat{Y} \in [0, 1]$. We repeated each experiment (for given data split ratio, oversampling-undersampling method, dataset, etc.) 30 times and summarized results over these repeats.

### 4.4 Evaluating Model Selection Approaches

Recall that *ROCCH method* selects a point on the convex hull; however, discrete classifiers from different classifiers may lie on the same point in ROC space. In such cases, we broke ties by selecting the discrete classifier which minimized cost among all discrete classifiers with the same (*FPR*, *TPR*)-coordinates. This is required because though these discrete classifiers have the same $FPR$ and $TPR$ on the validation set, they may have differing $FPR$ and $TPR$ on the test set. In short, if the ROCCH method correctly located the $(FPR, TPR)$ point on which the actual lowest test cost classifier lies, we considered it a correct model selection. The other approaches directly select a discrete classifier, and if two or more discrete classifiers are optimal and have the same metric value, then one is chosen at random.

We used three metrics to measure the empirical performance of the model selection approaches on the changed environment test sets. The primary metric was normalized cost (Provost & Fawcett, 2001) (Equation 2). We also explored using expected cost to measure performance since it incorporates both class and cost distribution information explicitly into its calculation (Equation 1). We also measured the Euclidean distance between selected discrete classifiers and the *Oracle* in ROC space $(FPR_{Oracle}, TPR_{Oracle})$. Given that for any value of $(FPR, TPR)$, $0 \le FPR \le 1$, and $0 \le TPR \le 1$, the distance is bounded as $0 \le \mathcal{D} \le \sqrt{2}$, with 0 being a perfect selection.

## 5 Results

We present a set of representative results here, with a more complete set of experimental settings reported in the appendix. As we varied the proportions of training, validation, and testing data, we observed the unsurprising result that, all else equal, smaller training set sizes produced higher costs. Therefore, we present results here on splits with 40% training data, 20% validation data, and 40% test data. We also observed that using *Random* versus *SMOTE-NearMiss* for introducing label shift did not substantially impact the results. Thus, we report here on experiments with the simpler process of random oversampling and undersampling for varying the class distributions. Another general trend we observe throughout the experiments is that matching the model selection metric to the test evaluation metric produces the highest quality model selections. In other words, *Norm-Cost-Min* performs best when evaluating with normalized cost, and *Exp-Cost-Min* performs best when evaluating with expected cost. We focus on here results based on the normalized cost since it is explicitly intended for empirical evaluation on test sets (Provost & Fawcett, 2001; Bettinger, 2003) and provide the expected cost results in the appendix (Section A.6). Finally, we also leave the simulated data results to the appendix (Section A.8.2), since the trends observed on the UCI datasets provided similar insights.

When we hold mislabeling costs constant (and equal to 1) and vary the class distribution while evaluating test performance with normalized cost, *Norm-Cost-Min* consistently performs best (closest to *Oracle*) among candidate model selection approaches on almost all datasets (Figure 2). *Accuracy-Max* suffers from inconsistency because it is unable to factor in the unequal class distributions. Interestingly, *F1-score-Max* is less consistent than *Accuracy-Max* on some datasets (`german-credit`, `climate-model-crashes`, etc. in Figure 2). *Exp-Cost-Min* optimizes expected cost, which is inconsistent with the test evaluation metric of normalized cost in Figure 2). When there is no label shift ($\frac{P_{test}(Y=1)}{P_{train}(Y=1)} = 1$), the *ROCCH method* is outperformed by other approaches on almost all datasets (Figure 2). Its model selection performance is often worst when the relative frequency of positives in the test set is higher than it was in the training set ($\frac{P_{test}(Y=1)}{P_{train}(Y=1)} > 1$). However, when the relative frequency of positives in test set is lower than in the training set ($\frac{P_{test}(Y=1)}{P_{train}(Y=1)} < 1$), the *ROCCH method* is sometimes competitive with *Norm-Cost-Min*. We also compared model selection approaches in terms of distance to *Oracle* in ROC space; these findings are in the appendix (Figure 7) and confirm the trends described above.

When we hold class distributions equal and vary the cost distributions while evaluating test performance with normalized cost, *Norm-Cost-Min* seems to be the most consistently best performing approach again (Figure 3). This is intuitive since *Norm-Cost-Min* explicitly incorporates the cost distribution of the changed test environment into its calculation (but does not depend on the class distribution of the test environment; Equation 2). *Accuracy-Max* and *F1-score-Max* perform poorly under skewed cost distributions (*e.g.*, $C(\hat{Y} = 0|Y = 1) = 3.0$), because they can not factor in the asymmetric costs. In Figure 3, the efficacy of the *ROCCH method* sometimes increases as the cost distributions become more skewed. For example, when $C(\hat{Y} = 0|Y = 1) = 1.0$, the *ROCCH method* does not perform well on `heart-disease`, `german-credit`, and `parkinsons`, but performs much better on those datasets when $C(\hat{Y} = 0|Y = 1) = 3.0$. Similar trends are apparent when measuring the Euclidean distance of the selected classifier from the *Oracle* in ROC space (appendix Figure 8)).

## 6 Discussion

The most interesting and perhaps surprising take-away from our experiments is this: simply choosing the classifier that optimizes performance on a validation set (weighted by available test condition information) is often empirically better than model selection via the *ROCCH method*, despite its theoretical grounding. This shows that the ROCCH method's assumption that the minimum cost classifier lies on the convex hull where the corresponding iso-performance line intersects it does not hold in most empirical settings; the actual minimum cost classifier lies elsewhere on the hull. Certainly, there are cases where the *ROCCH method* performs well; in particular, when the relative frequency of positives in the test set is lower than in the training set and when cost distributions are substantially skewed the *ROCCH method* can be competitive. However, even in those cases, *Norm-Cost-Min* (and *Exp-Cost-Min*, when measuring expected instead of normalized

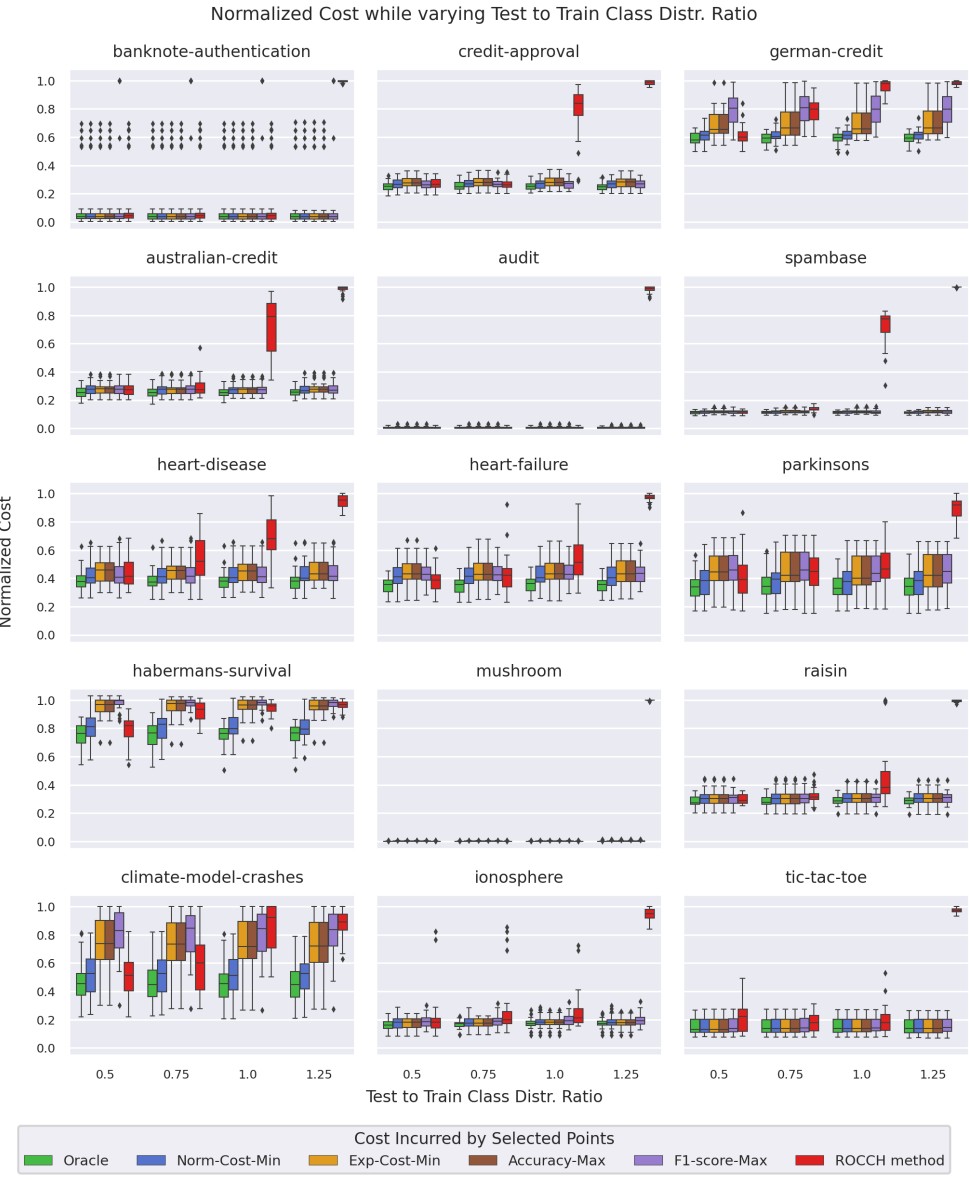

Figure 2: Effects of changing class distributions on model selection performance for 15 UCI datasets. Performance is quantified with normalized cost. *Oracle* shows the cost of the best classifier; this is the target for the model selection methods. *Norm-Cost-Min* performs best on almost all datasets. The *ROCCH method* struggles for class distribution ratios $\geq 1$ but works better when the test set has fewer positives than the training set.

cost) perform better, and in almost all other cases the *ROCCH method* does not perform competitively. Furthermore, ties created by the ROCCH method were broken in the most advantageous way to the method by picking the discrete classifier with best test performance, compared to using random selection for ties created by the other methods, and even then the ROCCH method underperformed. Therefore, this empirical study suggests using *Norm-Cost-Min* (or *Exp-Cost-Min*) over *ROCCH method* to select models for changed environments.

As noted above, the metric to be optimized on the validation set has substantial impact on the quality of the model selection process. Some metrics explicitly incorporate changed environment conditions (e.g,

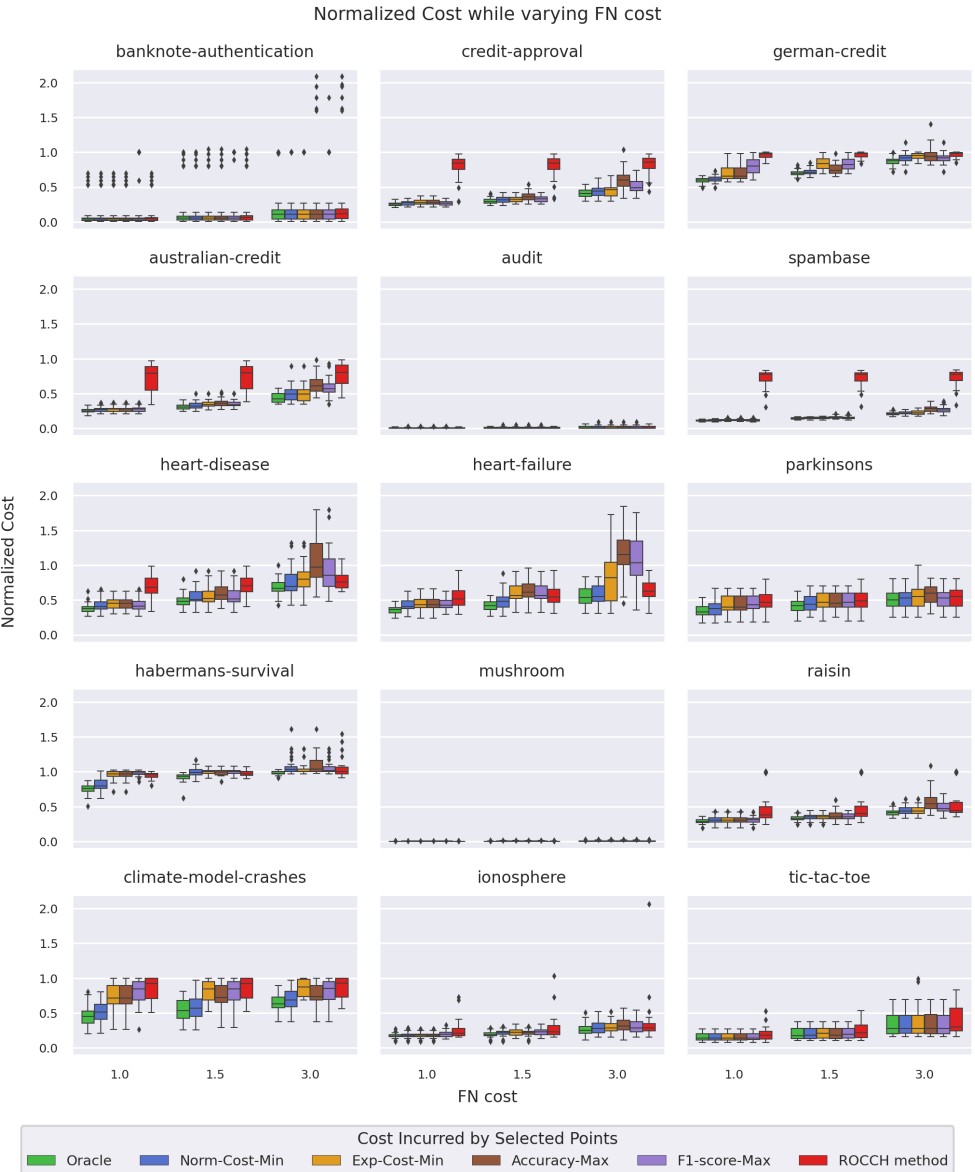

Figure 3: Effects of changing cost distributions on model selection performance for 15 UCI datasets. Performance is quantified with normalized cost. *Oracle* shows the cost of the best classifier; this is the target for the model selection methods. Methods other than *Norm-Cost-Min* suffer when cost distributions get more skewed.

normalized cost and expected cost), while others do not (e.g., accuracy and F1-score). Given that the datasets we have explored have asymmetric class distributions in the first place (Table 2 and 3), and that we have asymmetric cost distributions, metrics such as normalized cost and expected cost are better choices for evaluating selected models on the test set, rather than accuracy or F1-score. Furthermore, our results indicate that the same metric should be used on both the validation and testing sets.

One question we considered when interpreting the results of our experiments surrounded the degree to which covariate shift (assuming that $P_{train}(X) \neq P_{test}(X)$ while $P_{train}(Y|X) = P_{test}(Y|X)$ holds) may have played a role. In particular, we wondered whether our processes for generating changed test environments produced covariate shift, which might in turn impact our results. We explored several ways to measure covariate shift

between the training and test data and looked at how these measures related to the performance of the model selection approaches. We report on this investigation in the appendix, but for now, there are no conclusive trends to report (Section A.5).

Our study has a few potential limitations.

• **Positive Class Assignment.** Most of the UCI datasets have clearly defined positive and negative classes, though in some cases we have taken the liberty to make this assignment. For example, the `raisin` dataset has classes for two different species of raisins. Deciding which class to assign as positive is not as straightforward here as compared for example to the `heart-disease` dataset where we designate instances of the 'disease' class as positives and instances of the 'non-disease' class as negatives.

• **Artificially Created Changed Environments.** We created artificially changed test environments by manipulating class and cost distributions. Real world systems might present more challenging changed environments. For example, a changed environment might represent a time in the future, a different spatial region, or come with constraints specifying misclassifaction costs, and might present more complex types of data shift.

• **Experiment Design Choices.** We have explored many experimental settings, but there is always room for more detailed investigation. For example, explorations over finer granularity as well as wider ranges of class distribution and cost distribution could be possible, though we would expect to see similar trends. Additionally, one could imagine a similar exploration in a cross-validation framework instead of fixed training, validation, and testing splits.

## 7 Related Work

**Genesis.** Model selection of discrete classifiers for changed environments was first addressed by the ROCCH method (Provost & Fawcett, 1997; 1998; 2001). Selecting optimal classifiers for changed cost distributions has been studied independently by Adams & Hand (1999), through the lens of their proposed alternative to the ROC plot. We limit our study to model selection approaches based on ROC analysis.

**ROC Analysis.** The shortcomings of accuracy as a performance measure under class imbalance and/or unequal cost distributions (Provost et al., 1998) has led to the adoption of ROC analysis as a ubiquitous system of tools for evaluating machine learning models (Fawcett, 2006). ROC analysis (Fawcett, 2006; Flach, 2016b) provides a suite of multipurpose tools for different machine learning evaluation tasks, such as model visualization (using the ROC curve), model evaluation (using area under the ROC curve), model construction/hybridization (using the ROC convex hull), and model selection (using the ROCCH method) (Flach, 2004).

**Alternative Formulations.** Olecka (2002) introduced a linear programming framework which reformulates the ROCCH method as an optimization problem, using class and cost distributions of changed environments for defining their objective function. Lim & Won (2012) reformulate the ROC Convex Hull (the calculation of which is a important intermediate step of our candidate approaches) as a nonparametric maximum likelihood estimator of a ROC curve. We limit our scope to the original formulation by Provost & Fawcett (1997; 1998; 2001).

**Calibration.** Classifier calibration involves applying a post-processing step to scores predicted by continuous classifiers, with the intent to express them on a more meaningful scale (i.e., more closely resembling true class probabilities) (Flach, 2016a). Isotonic regression using the Pool Adjacent Violators (PAV) algorithm is a popular method for calibrating classifiers. Though the PAV and ROCCH methods were developed independently with non-overlapping intended uses, there exists a fundamental relationship between the PAV and ROCCH method, such that generating the ROC convex hull of a single continuous (score predicting) classifier is equivalent to isotonic regression using the PAV (Fawcett & Niculescu-Mizil, 2007). Individual hull segments on a ROC convex hull uniquely correspond to sets of pools in the PAV algorithm. Consequently, the ROC convex hull has been applied for the purpose of calibrating classifier scores (Bahnsen et al., 2014).

**Precision-Recall Analysis.** The confusion matrix or contingency table is the source of numerous machine learning metrics such as true positive rate/recall, false positive rate, precision, etc. Consequently, there exist

relationships between these metrics (and aggregations/combinations of these metrics). One such relationship of interest, is the one-to-one mapping between points in ROC space and points in Precision-Recall (PR) space, given non-zero recall (Davis & Goadrich, 2006). Furthermore, the analogous entity to the ROC convex hull in PR space is the achievable PR curve, and always contains the same points which constitute the ROC convex hull (Davis & Goadrich, 2006; Flach & Kull, 2015).

**Isometrics.** The ROCCH method uses the concept of isometrics, but this is a more general concept adopted in other theoretical research. The main idea behind isometrics is that some metrics have identical performance over identifiable regions in some space (e.g., ROC space, PR space). Flach (2003) formulate a 3-dimensional ROC space where the $x$, $y$ and $z$-axes are false positive rate, true positive rate, and class distribution, respectively. They show that, for a given performance metric (such as accuracy) and a given value of that metric (e.g., 80%), all classifiers which achieve this value for this metric lie on the same 'isosurface' in 3D ROC space. They analytically study the projections of these 3-D isosurfaces back on to 2-D ROC space, to investigate and understand how metrics such as accuracy, precision, F-measure, information gain, Gini split, etc., generally behave with respect to changing performance and class distributions.

**Cost, Threshold Choice, and Model Selection.** Model selection as a threshold choice problem based on expected loss/cost has been studied by Hernández-Orallo et al. (2012). They show that calibration plays an important role in the process of threshold (discrete model) selection. Our work investigates model selection approaches for classifiers that are already on the ROC convex hull, which represents a PAV-calibrated classifier itself (Fawcett & Niculescu-Mizil, 2007)).

**Critique on Model Selection Performance.** There has been some back and forth correspondence between the authors and critics of the ROCCH method. Webb & Ting (2005) raise concerns about use of the method under varying class distributions. They argue that the response variable might be causally dependent on certain features or feature subspaces, in which case it is not possible for the method to make consistent model selections. This claim was also supported by the authors in their response to the critique (Fawcett & Flach, 2005). However, Fawcett & Flach (2005) also remind us that such deterministic situations might not be common in the real world, and that the ROCCH method still holds promise.

**Causality, Domain Adaptation, and Counterfactual Estimation.** The authors of the ROCCH method adopt a discriminative view while formulating the problem setting of changed environments, which we continue in our work. Fawcett & Flach (2005) acknowledge that an alternative way of formulating the problem can be motivated by causality, which adopts a generative view. Domain adaptation has also been studied through the lens of causality (Zhang et al., 2015). Counterfactual estimation can improve performance under domain shift (Bottou et al., 2013); this approach is motivated by causality and also relies on generative modeling. It is entirely possible that one of more classes are causally dependent on certain feature subgroups which are not effectively represented during testing, but as mentioned previously we limit our scope to the original discriminative view adopted by the authors of the ROCCH method.

## 8 Future Work

**Calibration.** Though the ROC convex hull represents a PAV calibrated classifier, there are other calibration methods (such as Platt scaling) which may be better suited (Hernández-Orallo et al., 2012). Currently we do not explicitly calibrate predicted class probabilities. A portion of the data would need to be used for calibration because explicitly calibrating continuous classifiers requires having separate calibration datasets. Explicit calibration of continuous classifiers would help us understand how different calibration methods such as PAV, Platt scaling, etc. affect model selection.

**Label Shift and Asymmetric Trends.** The ROCCH method seems to show asymmetric behaviour for cases when we oversample negatives compared to when we oversample positives. Further investigation is required to reveal if this is a property of the method, the datasets, the experimental settings, or some combination of these.

**Mismatch between Expected and Actual Testing Conditions.** We only explore experimental settings when the information about testing conditions provided to the model selection methods were correct and

exact. However, it would be interesting to study the tolerance of these model selection methods to incorrect environmental conditions, i.e., when expected testing conditions do not match actual testing conditions.

**Real-World Applications.** Applying to more real world datasets seems to be the most important next step for investigating the efficacy of model selection methods under changed environments, especially with clearly defined changed environment test datasets. Some examples of such changed environments in the context of species distribution modeling might be - (i) *changing climatic conditions*, where changing temperatures may present covariate shift and label shift if species prevalence is affected, (ii) *different spatial regions*, where covariate shift and label shift can also arise when training and testing data are spatially disparate, and (iii) *different misclassification costs*, where false negatives may be more costly than false positives for rare species or to evaluate certain management proposals.

## 9 Conclusions

When approaching a discrete classification problem for which the anticipated testing environment differs in class and/or cost distribution from the training data at hand, two main recommendations can be gleaned from the investigation reported above. First, it is prudent to choose a performance metric that can explicitly incorporate properties of the changed test environment; expected cost meets this requirement, but accuracy and F1-score do not. Second, a simple procedure that weights validation set costs by the anticipated class and/or cost distributions of the test set has good empirical performance across a range of datasets and environments. Our experiments suggest to use this procedure over the ROCCH method, despite its theoretical grounding.

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

# A  Appendix

## A.1  Glossary of Terms

---

**Class Distribution**: relative frequency of positives $P(Y = 1)$ and negatives $P(Y = 0)$ where $P(Y = 1) = 1 - P(Y = 0)$.

**Cost Distribution**: false positive cost $C(\hat{Y} = 1|Y = 0)$ and false negative cost $C(\hat{Y} = 0|Y = 1)$.

**Changed Environments**: represented by test datasets that have different class and/or cost distributions compared to training datasets. We denote label shift using test class distribution to train class distribution ratio $\frac{P_{test}(Y=1)}{P_{train}(Y=1)}$.

**Continuous Classifiers**: score-predicting classifiers, i.e., $\hat{Y} \in \mathbb{R}$.

**Discrete Classifiers**: discrete label predicting classifiers, i.e., $\hat{Y} \in \{0, 1\}$.

**ROC Space**: composed of $FPR$ on the $x$-axis and $TPR$ on the $y$-axis. We can obtain the numbers of false positives ($FP$), true negatives ($TN$), true positives ($TP$) and false negatives ($FN$) from a confusion matrix. Coordinates of points in ROC can then be calucates as follows $FPR = FP/(FP + TN)$ and $TPR = TP/(TP + FN)$.

**ROC Curves**: curves in ROC space composed of ($FPR$, $TPR$) points (Fawcett, 2006).

**ROC Convex Hull**: an ROC curve with ($FPR, TPR$) points such that those points form the convex hull of all points (discrete classifiers) from all continuous classifiers (Provost & Fawcett, 2001).

**Iso-Performance Line**: abstract linear regions in ROC space where classifiers have equal performance.

**ROCCH method**: an approach of model selection for changed environments. Provost & Fawcett (1997; 1998; 2001) address model selection for changed environments by introducing the ROCCH method.

**Expected Cost**: for a given discrete classifier yielding a ($FPR$, $TPR$) point, the expected cost (Provost & Fawcett, 2001) of making a classification is,

$$P(Y = 1) \cdot (1 - TPR) \cdot C(\hat{Y} = 0|Y = 1) + P(Y = 0) \cdot FPR \cdot C(\hat{Y} = 1|Y = 0)$$

**Normalized Cost**: refers to normalized expected cost (Provost & Fawcett, 2001).

$$(1 - TPR) \cdot C(\hat{Y} = 0|Y = 1) + FPR \cdot C(\hat{Y} = 1|Y = 0)$$

---

## A.2  Derivation of the Slope of the Iso-Performance Line for Expected Cost

Let us consider two points in ROC space $(FPR_1, TPR_1)$ and $(FPR_2, TPR_2)$ which incur expected costs $\mathcal{C}_1$ and $\mathcal{C}_2$ respectively.

$$\mathcal{C}_1 = P(Y = 1) \cdot (1 - TPR_1) \cdot C(\hat{Y} = 0|Y = 1) + P(Y = 0) \cdot FPR_1 \cdot C(\hat{Y} = 1|Y = 0)$$

$$\mathcal{C}_2 = P(Y = 1) \cdot (1 - TPR_2) \cdot C(\hat{Y} = 0|Y = 1) + P(Y = 0) \cdot FPR_2 \cdot C(\hat{Y} = 1|Y = 0)$$

When $\mathcal{C}_1 = \mathcal{C}_2$ (these two points incur equal expected costs), we get

$$P(Y = 1) \cdot (1 - TPR_1) \cdot C(\hat{Y} = 0|Y = 1) + P(Y = 0) \cdot FPR_1 \cdot C(\hat{Y} = 1|Y = 0) =$$
$$P(Y = 1) \cdot (1 - TPR_2) \cdot C(\hat{Y} = 0|Y = 1) + P(Y = 0) \cdot FPR_2 \cdot C(\hat{Y} = 1|Y = 0)$$
$$\Rightarrow P(Y = 1) \cdot (1 - TPR_1) \cdot C(\hat{Y} = 0|Y = 1) - P(Y = 1) \cdot (1 - TPR_2) \cdot C(\hat{Y} = 0|Y = 1) =$$
$$P(Y = 0) \cdot FPR_2 \cdot C(\hat{Y} = 1|Y = 0) - P(Y = 0) \cdot FPR_1 \cdot C(\hat{Y} = 1|Y = 0)$$
$$\Rightarrow P(Y = 1) \cdot C(\hat{Y} = 0|Y = 1) \cdot ((1 - TPR_1) - (1 - TPR_2)) =$$
$$P(Y = 0) \cdot C(\hat{Y} = 1|Y = 0) \cdot (FPR_2 - FPR_1)$$
$$\Rightarrow P(Y = 1) \cdot C(\hat{Y} = 0|Y = 1) \cdot (1 - TPR_1 - 1 + TPR_2) =$$
$$P(Y = 0) \cdot C(\hat{Y} = 1|Y = 0) \cdot (FPR_2 - FPR_1)$$
$$\Rightarrow P(Y = 1) \cdot C(\hat{Y} = 0|Y = 1) \cdot (TPR_2 - TPR_1) =$$
$$P(Y = 0) \cdot C(\hat{Y} = 1|Y = 0) \cdot (FPR_2 - FPR_1)$$
$$\Rightarrow \frac{TPR_2 - TPR_1}{FPR_2 - FPR_1} =$$
$$\frac{P(Y = 0) \cdot C(\hat{Y} = 1|Y = 0)}{P(Y = 1) \cdot C(\hat{Y} = 0|Y = 1)}$$

The slope of the iso-performance line is this product of the reciprocal of class distribution ratio $\frac{P(Y=0)}{P(Y=1)}$ and cost distribution ratio $\frac{C(\hat{Y}=1|Y=0)}{C(\hat{Y}=0|Y=1)}$.

### A.3 Additional Details on Experimental Design

Figure 4 shows an overview of the methodology we used for our experiments. We split datasets in to training, validation, and testing datasets (Figure 5) according to specified relative sizes. We introduced label shift by changing class distribution of test datasets (Figure 6). Continuous classifiers are trained on the training set; this concludes the learning part. The trained continuous classifiers predict on the validation set, which were thresholded to obtain discrete labels, and corresponding discrete classifiers of the form $(F, T)$, where $F$ refers to the continuous classifier and $T$ refers to the value used to threshold predicted scores. ROC curves were constructed using these discrete labels followed by finding the ROC convex hull. The ROC curve of continuous classifier $F$ is composed of discrete classifiers $(F, T)$. Finally, we selected among discrete classifiers on the ROC convex hull, and evaluate against *Oracle*.

This entire process was repeated 30 times for each dataset with different folds of training, validation and testing data.

We vary our experimental settings as follows,

- *Train Ratio*: Amount of data to be used for training. Varied in the range $\{0.2, 0.4, 0.6\}$. *Train Ratio* $= 0.2$ implies that 20% of data is used for training classifiers.

- *Validation Ratio*: Amount of data to be used for validation. Varied in the range $\{0.2, 0.4, 0.6\}$.

- *Test Ratio*: Amount of data to be used for testing. Varied in the range $\{0.2, 0.4, 0.6\}$.

- *Oversamping-Undersampling Method*: Method for oversampling and undersampling test data. We either use *Random* or *SMOTE-NearMiss*.

- *Test to Train Class Distr. Ratio*: Class distribution shift presented by changed environment. Varied according in the range $\frac{P_{test}(Y=1)}{P_{train}(Y=1)} \in \{0.5, 0.75, 1.0, 1.25\}$.

- *FN Cost*: Cost of predicting a false negative. Varied in the range $C(\hat{Y} = 0|Y = 1) \in \{1.0, 1.5, 3.0\}$.

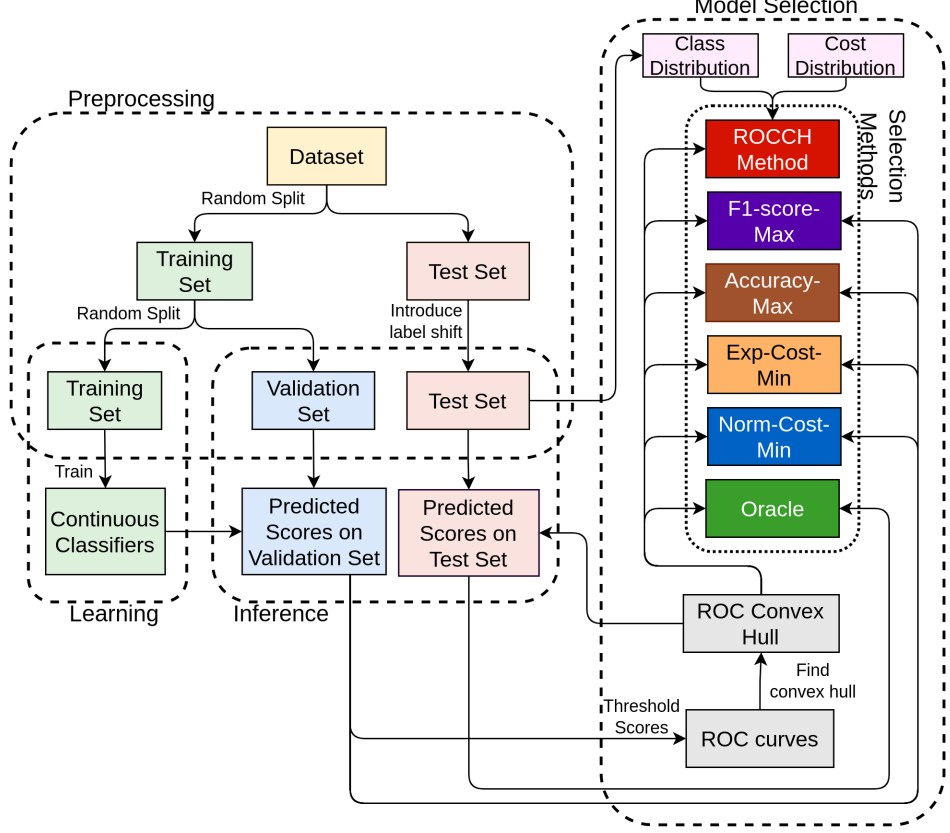

Figure 4: Our experimental methodology. Preprocessing: Datasets were randomly split into training, validation and test sets, and label shift is introduced. Learning: Continuous binary classifiers were trained. Inference: Learned continuous classifiers produce predicted scores on validation set and (modified) test set. Model Selection: Various model selection methods selected discrete binary classifiers on the convex hull, which were then evaluated on the test set. Connections from Cost Distribution to Norm-Cost-Min, and from Class Distribution and Cost Distribution to Exp-Cost-Min are not shown for clarity.

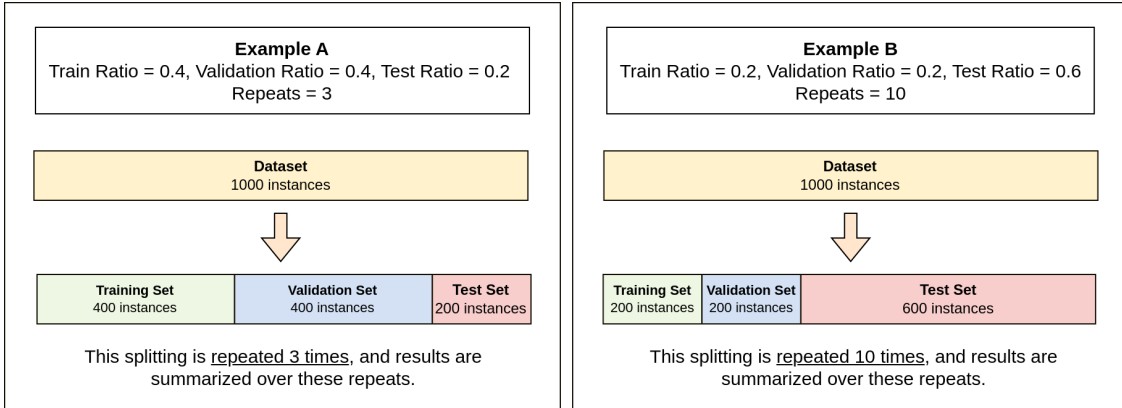

Figure 5: Our approach of splitting datasets in to training, validation and test sets. Example A and B represent two possible experimental settings.

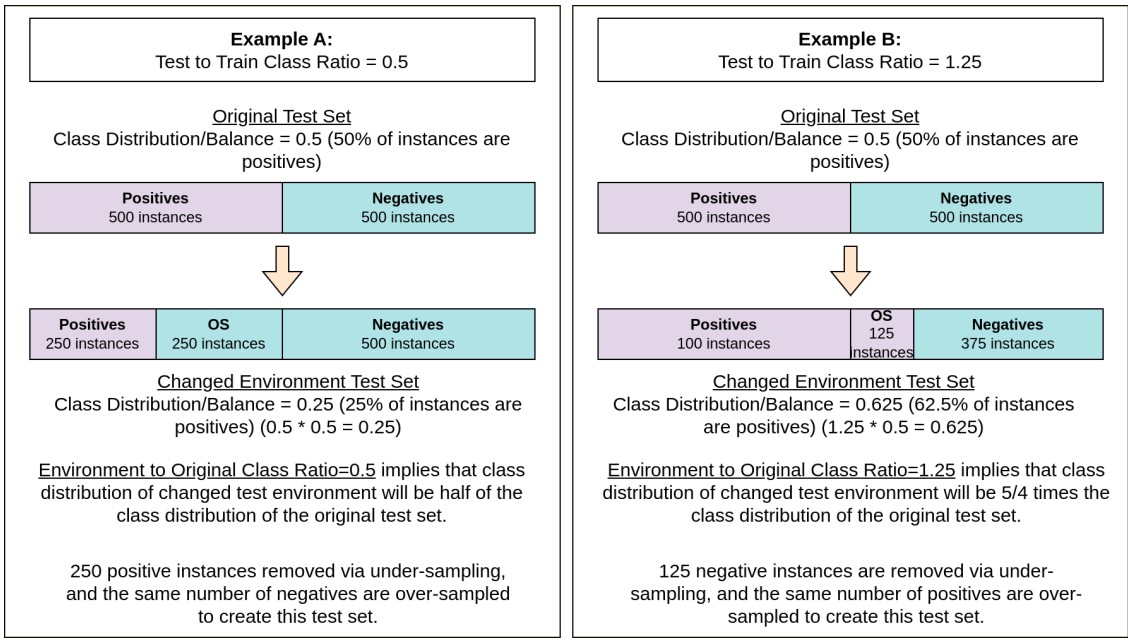

Figure 6: Our approach of varying class distribution of test set (introducing label shift). OS refers to oversampled instances. Example A and B represent two possible experimental settings.

## A.4 Results Based on Distance from Oracle

We observed that high cost classifiers also lie further away from *Oracle* in ROC space. Figure 7 shows that classifiers that incur low normalized cost on changed test environment data (Figure 2), also lie closer to *Oracle* in ROC space in terms of Euclidean distance.

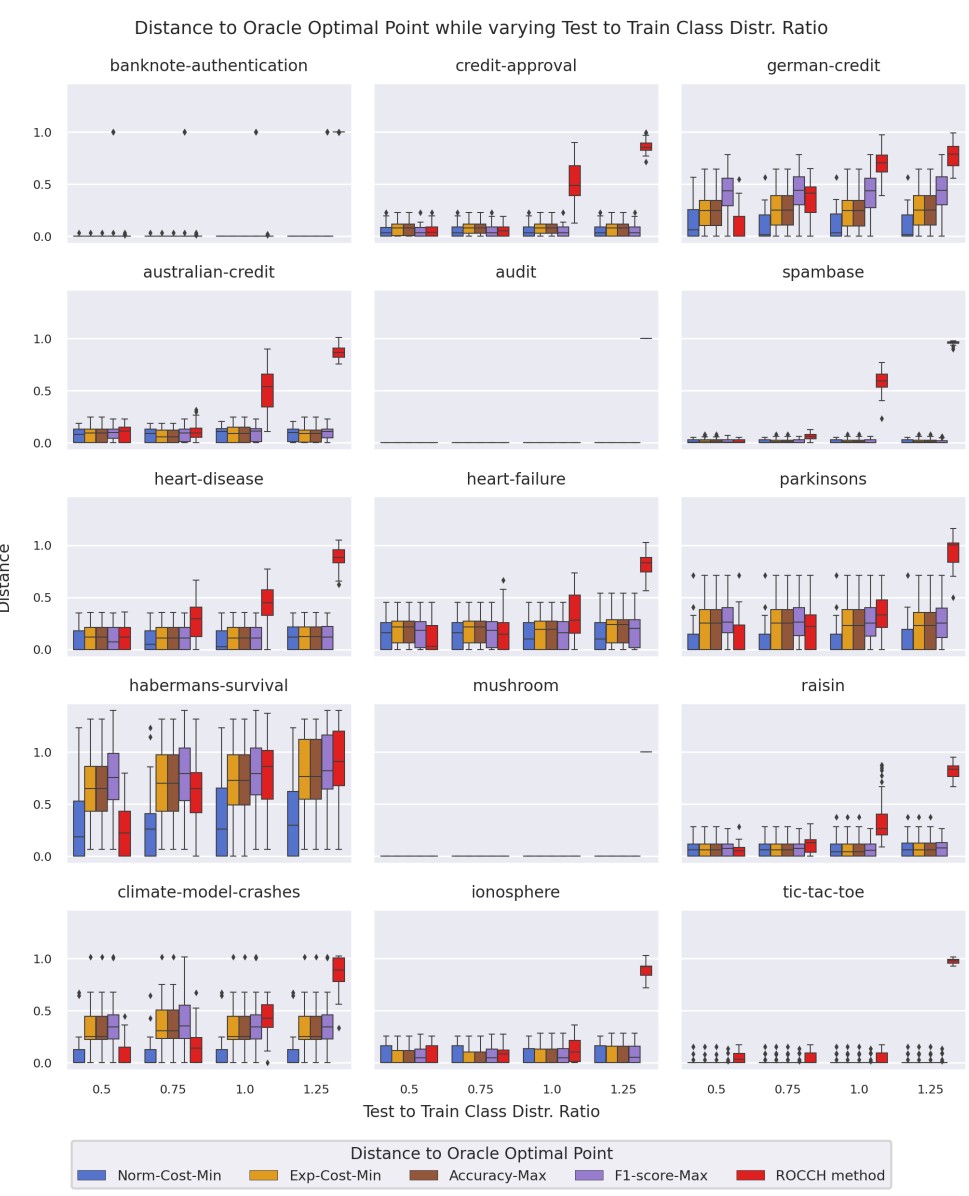

Figure 7: Effects of changing class distributions on model selection performance for 15 UCI datasets. Performance is quantified with distance from *Oracle*. Classifier selected by Norm-Cost-Min is generally closest to Oracle.

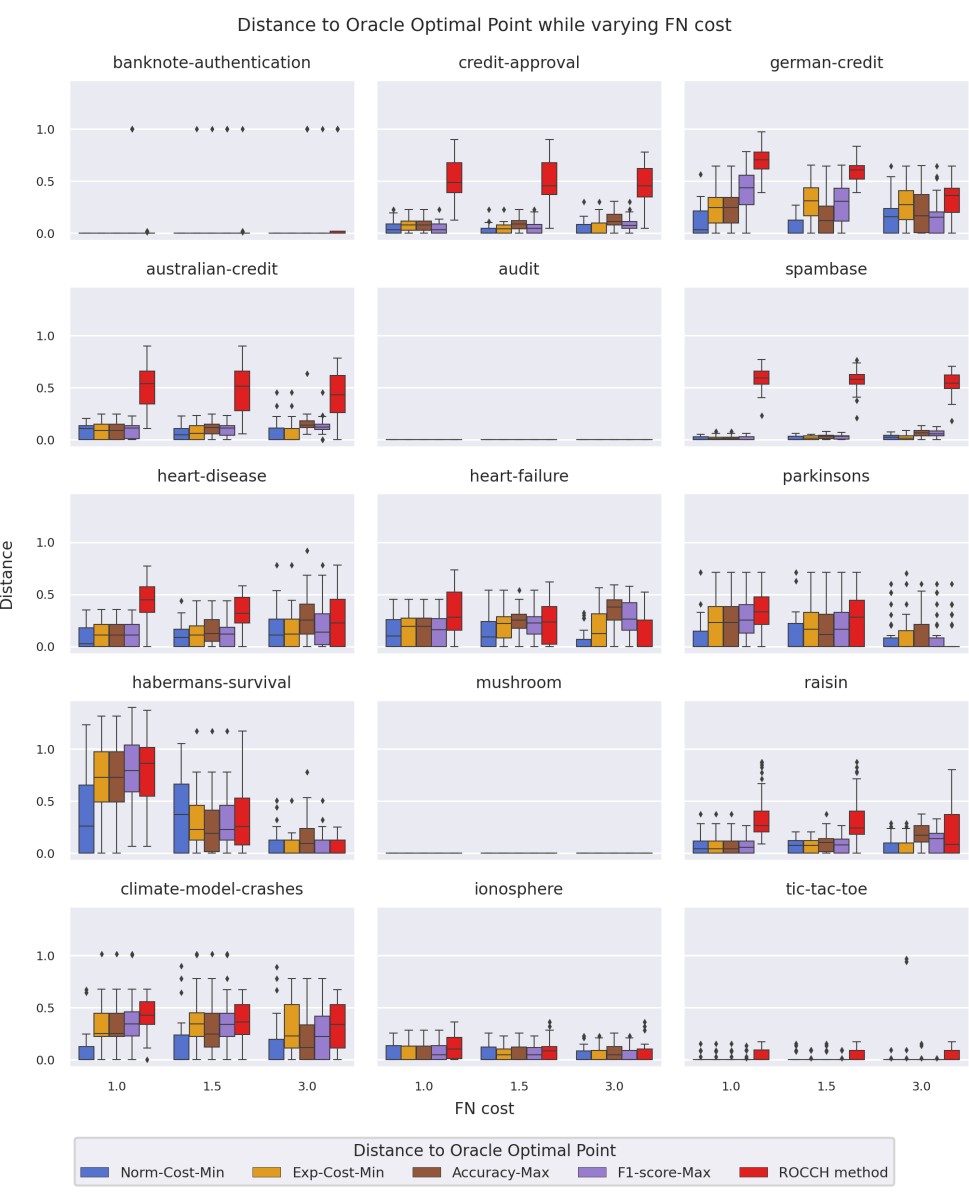

Figure 8: Effects of changing cost distributions on model selection performance for 15 UCI datasets. Performance is quantified with distance from *Oracle*.

### A.5 Covariate Shift

#### A.5.1 Covariate Shift Measurement

There might be intrinsic covariate shift between training and testing data (depending on average heterogeneity of instances within a dataset), which may be further affected when test data is subjected to label shift. We measure covariate shift via several approaches,

- *Wasserstein Distance*: Also known as the Earth Mover's Distance. Measures the distance between two distributions. We calculated for each feature and averaged over all features.

- *Energy Distance*: A metric for measuring statistical distance between two distributions. Similarly to *Wasserstein Distance*, we calculated for each feature, and then averaged over all features.

- *Maximum Mean Discrepancy (MMD)*: Measures distance between distributions by calculating the distance between "mean embeddings" of features. We used a linear kernel.

- *AUC*: A random forest classifier was trained to differentiate between instances of training and testing sets. Performance was measured by using 20% of data for evaluation, and calculating Area under the ROC (AUC). The process was repeated 3 times using random splits and averaged.

- *Matthews Correlation Coefficient (MCC)*: A discrete version of Pearson Correlation Coefficient. Same approach (evaluation set size, number of repeats) was used as for calculating the *AUC*.

- *Cramér-von Mises Criterion*: Compares two empirical distributions by measuring goodness of fit. Similar to *Wasserstein Distance* and *Energy Distance*, we calculated for each feature, and then averaged over all features.

#### A.5.2 Results on Covariate Shift and ROCCH Method

The *ROCCH method* behaves very differently when we oversample positives and undersample negatives compared to when we oversample negatives and undersample positives. We investigate how covariate shift affects model selection performance (shown in Figure 9). In Figure 9, rows of subfigures corresponding to different covariate shift measurement methods and columns of subfigures correspond to train to test class distribution ratios (label shift). It is evident that covariate shift measurement metrics show varying trends. *Wasserstein Distance* and *Energy Distance* show almost near identical results. Similarly, *AUC* and *MCC* also show near identical results. This is due to similarity between how methods measure covariate shift. Our calculation of *AUC* and *MCC* uses the same underlying framework.

One paradoxical behaviour we observe is that for conditions when the class and cost distributions of the test environment is same as for the training environment (we do not make any modifications nor use altered costs), all metrics are negatively correlated to error, though 4 out of the 6 metrics did not have significant *p*-values (third column of subfigures from the left in Figure 9). If we were to assume this weak negative correlation to be correct, it would indicate that more covariate shift implies better model selection, which is not intuitive. However, for $\frac{P_{test}(Y=1)}{P_{train}(Y=1)} < 1$, we see an expected behavior where the more covariate shift and error are positively correlated (we see this trend for all metrics except *Cramér-von Mises Criterion*). We present results for cases when we used *Random* oversamping and undersampling. Therefore, only copies of instances from the test set were added back in to the test set, and any covariate shift we observe is due to heterogeneity of instances in feature space within each dataset. This is an unavoidable artifact of splitting datasets in to training and test sets, which we hope to mitigate by stratification during random splitting, but we observe as covariate shift. We anticipate that comparing model selection efficacy against real-world covariate shift might reveal clearer trends.

### A.6 Results Based on Expected Cost

When we evaluate model selection performance based on expected cost, we observe that *Exp-Cost-Min* performs best under changing class distributions (Figure 10 and 11) and cost distributions (Figure 12 and

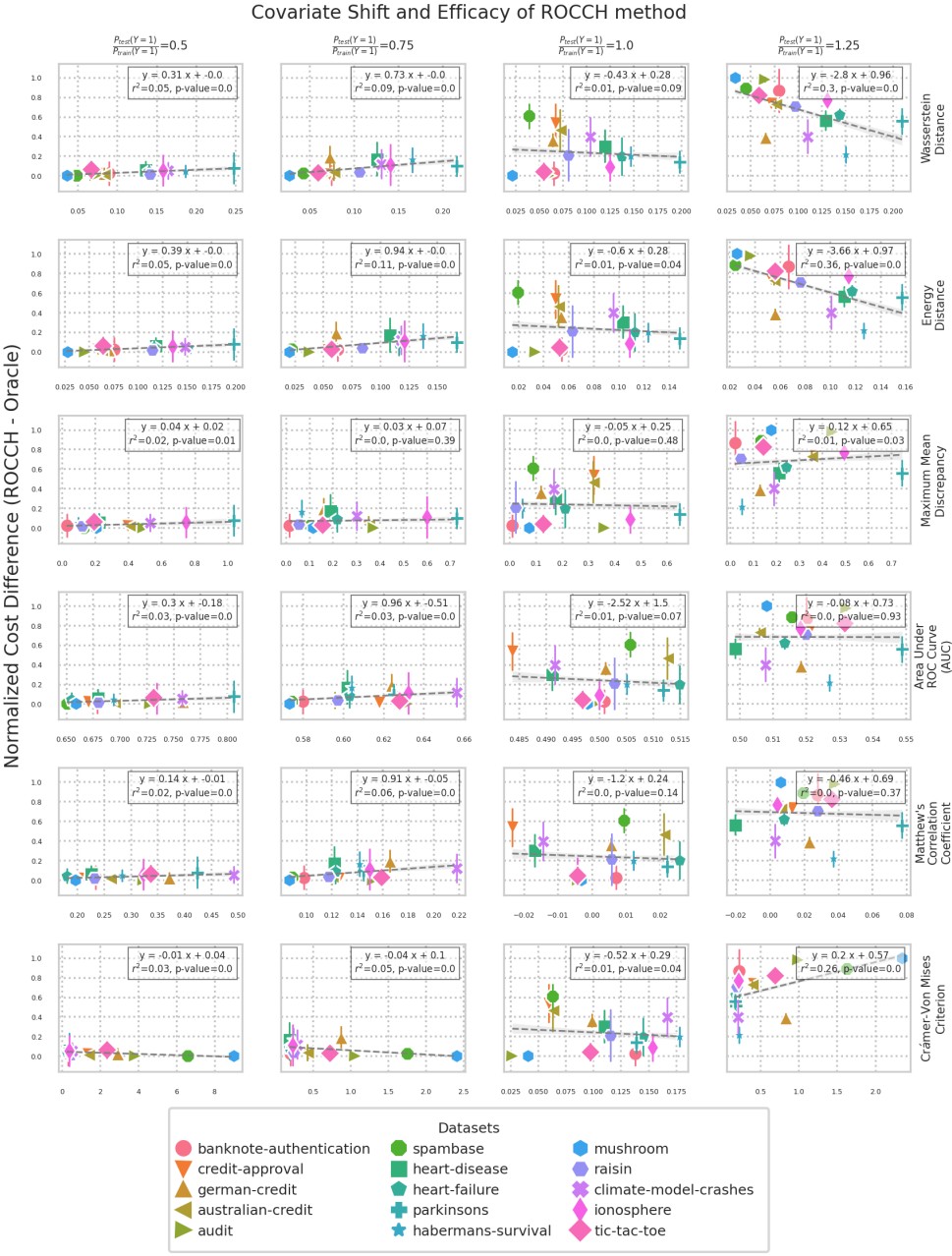

Figure 9: Effects of changing covariate shift on model selection performance, based on normalized cost, and applied to 15 UCI datasets. x-coordinates were derived by averaging covariate shift over all 30 runs. Metrics do not show any clear or conclusive behavior.

13). Our experiments for studying relationship between covariate shift and expected cost incurred by models selected by *ROCCH method* do not indicate any conclusive trends (Figure 14) (similar to normalized cost).

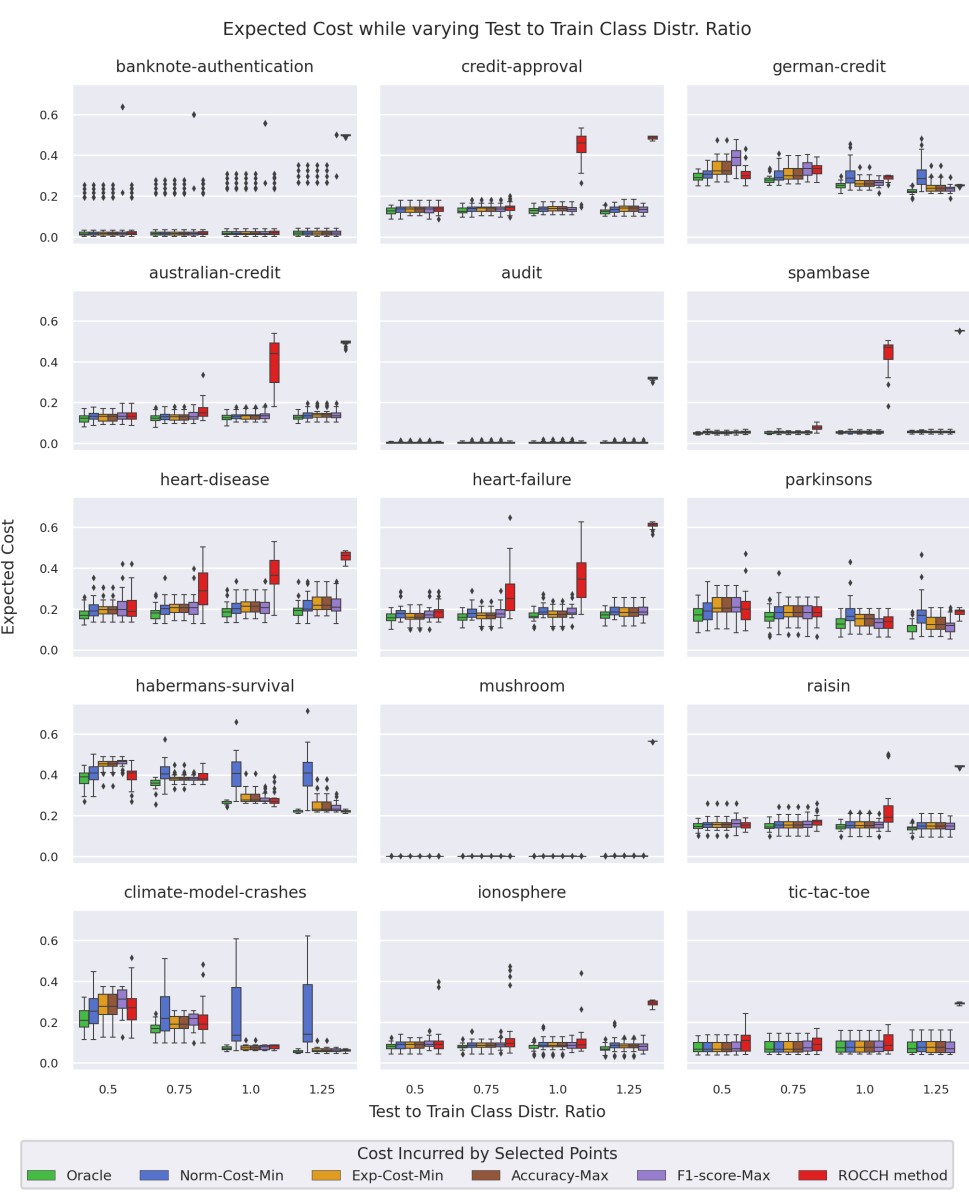

Figure 10: Effects of changing class distributions on model selection performance for 15 UCI datasets. Performance is quantified with expected cost. *Oracle* shows the cost of the best classifier; this is the target for the model selection methods.

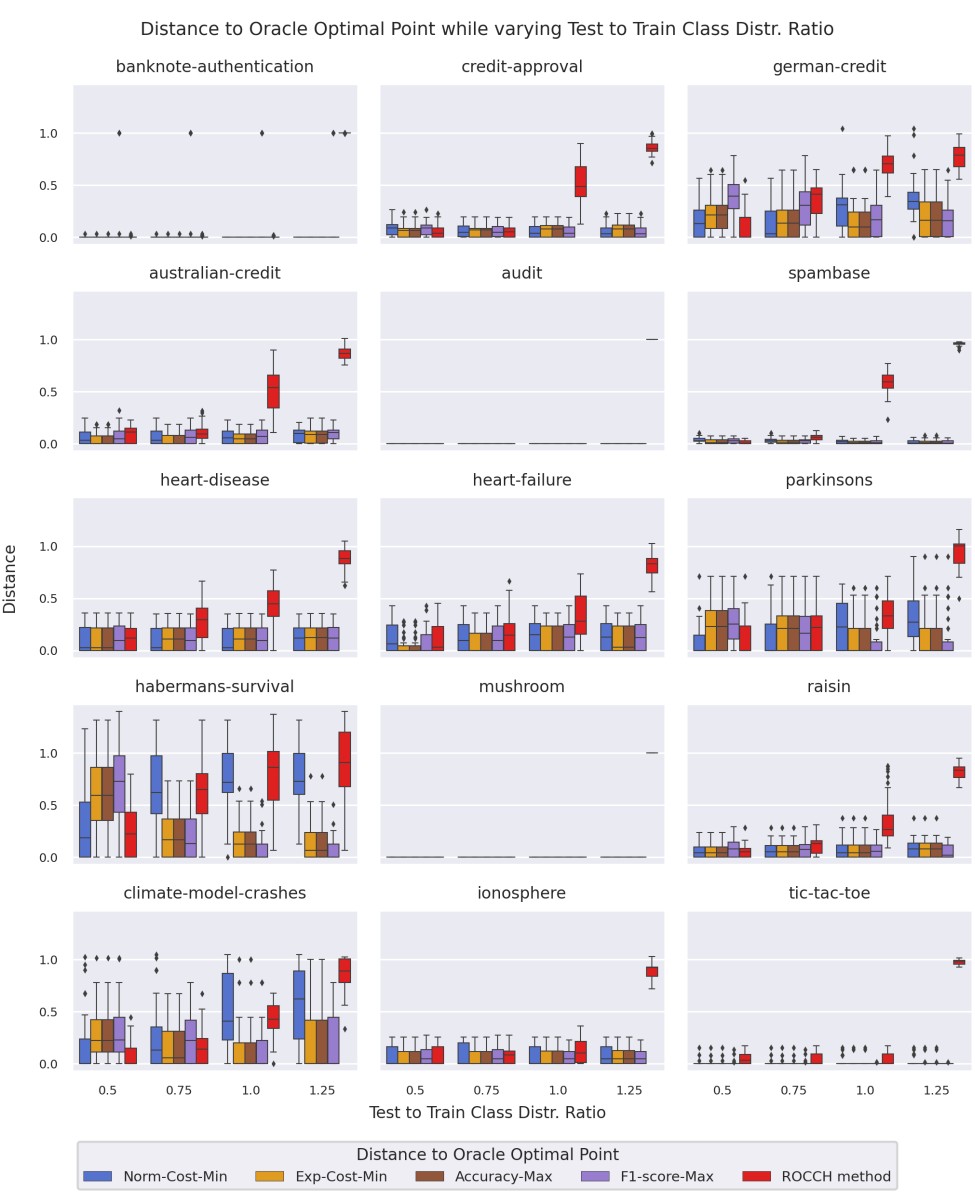

Figure 11: Effects of changing class distributions on model selection performance for 15 UCI datasets. Performance is quantified with distance from *Oracle*, based on expected cost.

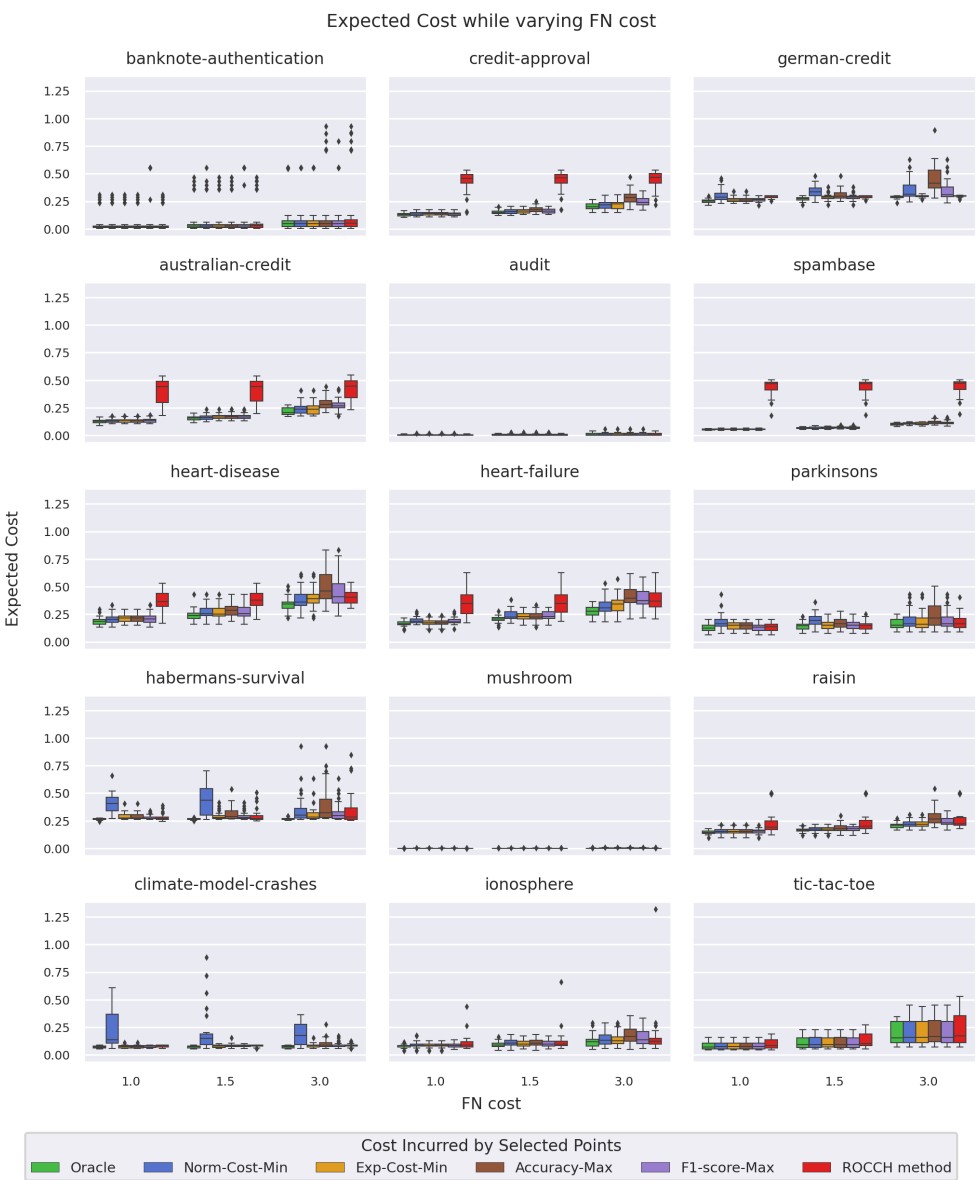

Figure 12: Effects of changing cost distributions on model selection performance for 15 UCI datasets. Performance is quantified with expected cost. *Oracle* shows the cost of the best classifier; this is the target for the model selection methods.

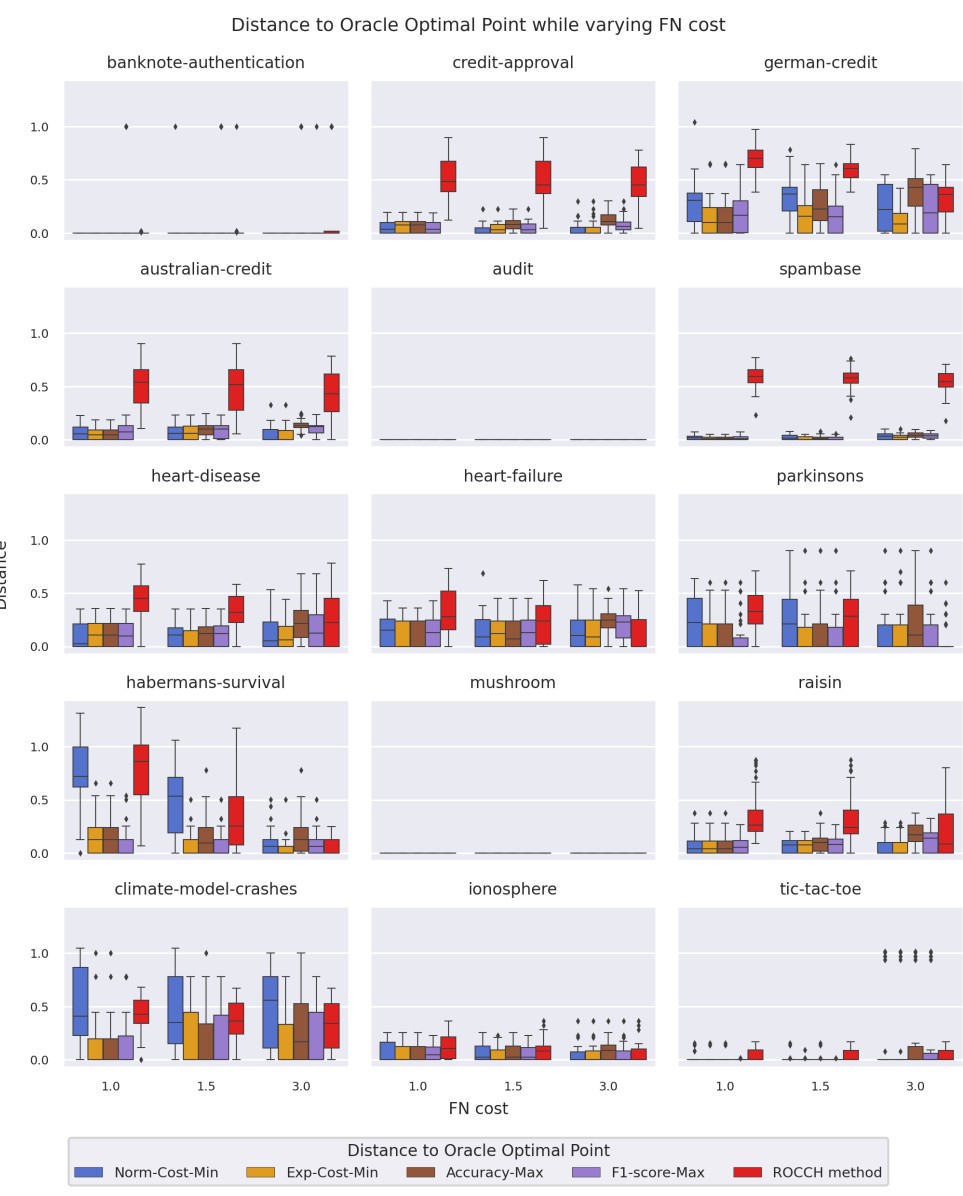

Figure 13: Effects of changing cost distributions on model selection performance for 15 UCI datasets. Performance is quantified with distance from *Oracle*, based on expected cost.

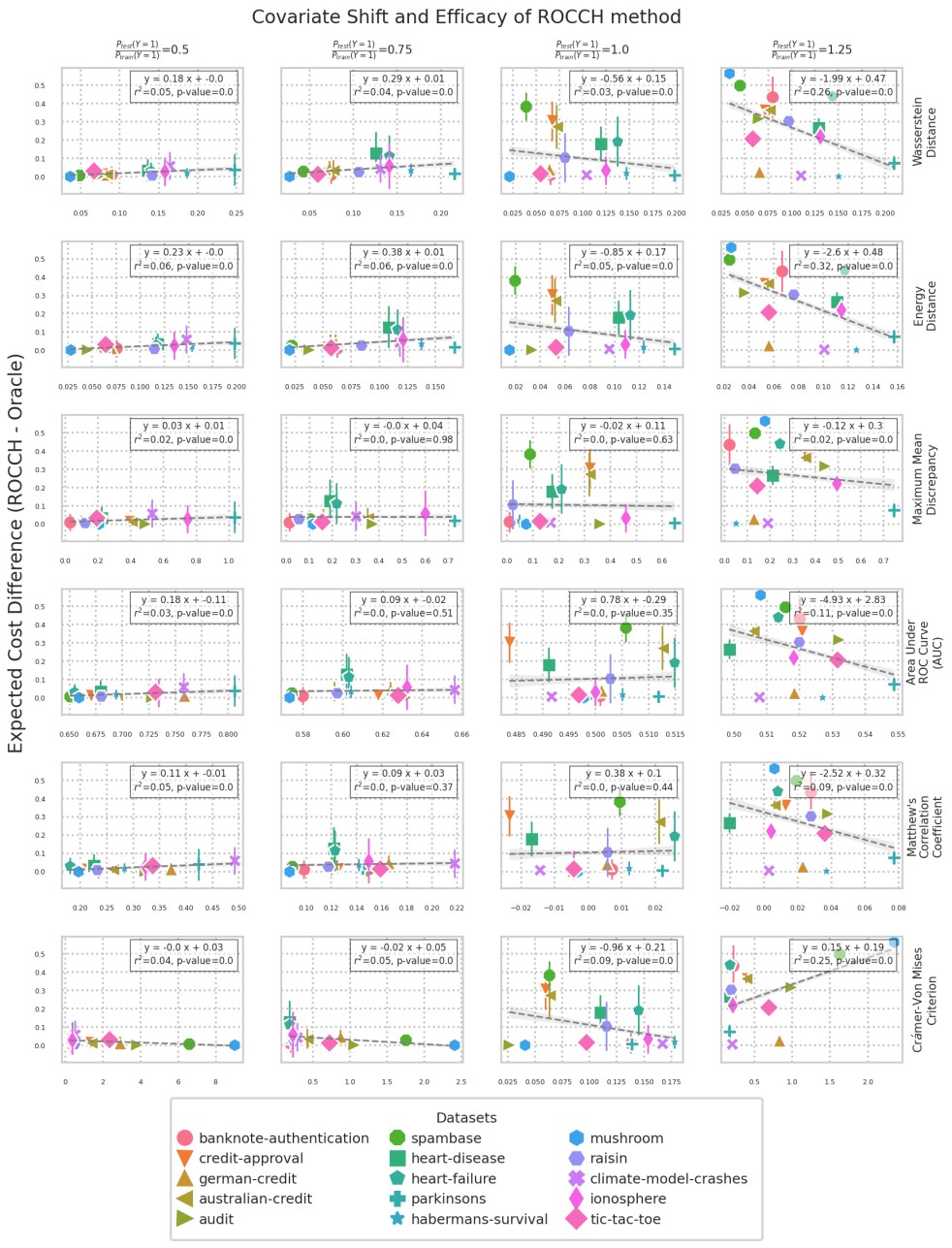

Figure 14: Effects of changing covariate shift on model selection performance, based on expected cost, and applied to 15 UCI datasets. x-coordinates were derived by averaging covariate shift over all 30 runs.

## A.7 Varying Dataset Characteristics

For a given row of subfigures in Figure 15, the x-coordinate of each dataset remains constant, only the y-coordinate varies. Again, we observe asymmetric behaviour when we compare cases when $\frac{P_{test}(Y=1)}{P_{train}(Y=1)} \geq 1$ and when $\frac{P_{test}(Y=1)}{P_{train}(Y=1)} < 1$. For example, dataset size and no. of features are positively correlated to error when $\frac{P_{test}(Y=1)}{P_{train}(Y=1)} \geq 1$, but negatively correlated to error when $\frac{P_{test}(Y=1)}{P_{train}(Y=1)} < 1$.

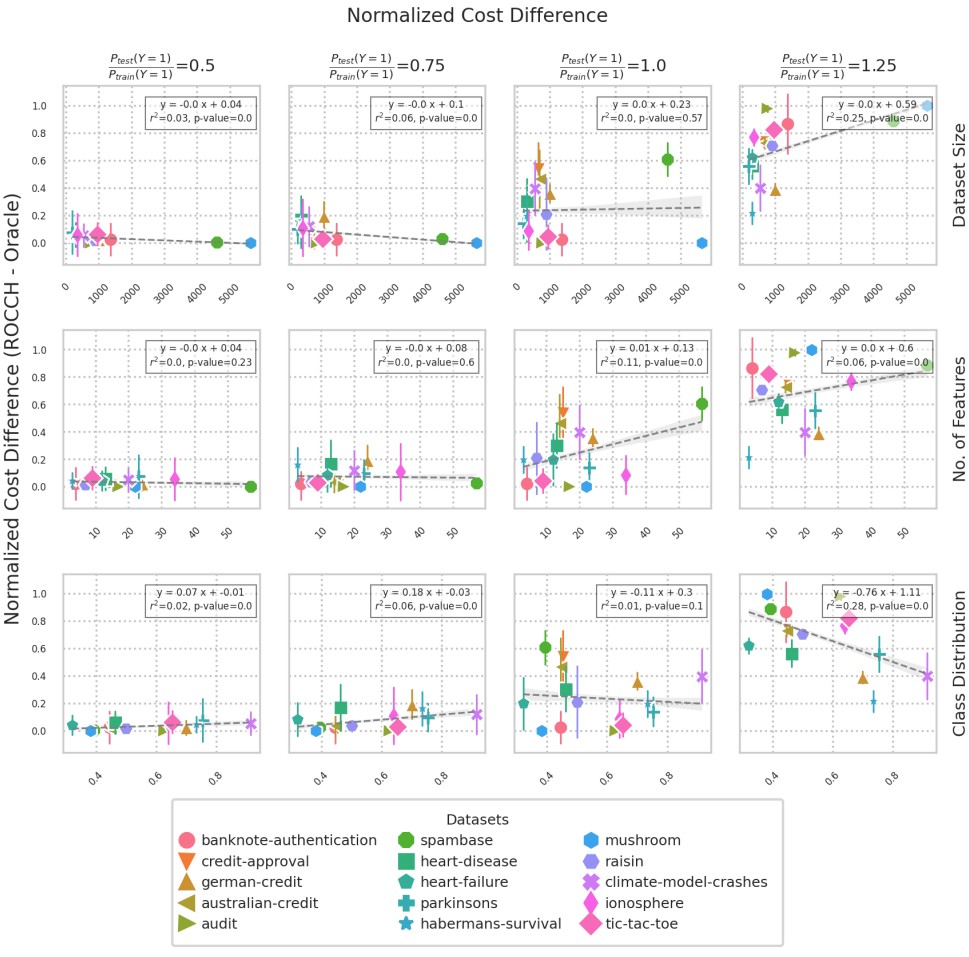

Figure 15: Effects of changing dataset characteristics on model selection performance for 15 UCI dataset. Performance is quantified with normalized cost.

### A.8 Simulated Data Analysis

### A.8.1 Data

For features $X$,

- *Continuous Features*: Drawn from standard normal distribution $X_i \sim N(0,1)$.

- *Categorical Features*: Drawn from Bernoulli distribution $X_i \sim Bern(0.5)$

$$z = \beta_0 + \beta_1 X_1 + \beta_2 X_2 + \beta_2 X_3 + \beta_4 X_4$$

$$p = \frac{1}{1 + exp(-z)}$$

$$Y = Bern(p)$$

We set $\beta_i = 1$ for $1 \le i \le 4$ and vary the intercept $\beta_0$ to vary the class distribution/balance. Intercept term is varied slightly to ensure that approximate class balance meets requirements.

We also add feature noise to some simulated datasets. Particularly, we add noise to 20% of randomly selected positive instances. Let us denote these selected positive instances as $X^{noisy}$. For continuous features, $X^{noisy} = X + e$ where $e \sim N(0,1)$. For categorical features $X^{noisy} = 1$ if $X = 0$, and $X^{noisy} = 0$ if $X = 1$, essentially flipping the bits.

Table 3 shows our generated simulated datasets, and their properties.

Table 3: Simulated datasets.

| No. | Dataset | Instances | Features | Categorical Features | Class Balance | Added Noise |
|---|---|---|---|---|---|---|
| 1 | dataset-1 | 500 | 4 | 0 | 0.5 | × |
| 2 | dataset-2 | 500 | 4 | 0 | 0.25 | × |
| 3 | dataset-3 | 1000 | 4 | 0 | 0.5 | × |
| 4 | dataset-4 | 1000 | 4 | 2 | 0.5 | × |
| 5 | dataset-5 | 1000 | 4 | 4 | 0.5 | × |
| 6 | dataset-6 | 1000 | 4 | 0 | 0.25 | × |
| 7 | dataset-7 | 1000 | 4 | 0 | 0.75 | × |
| 8 | dataset-8 | 1000 | 4 | 0 | 0.5 | ✓ |
| 9 | dataset-9 | 1000 | 4 | 0 | 0.25 | ✓ |
| 10 | dataset-10 | 1000 | 4 | 2 | 0.5 | ✓ |
| 11 | dataset-11 | 1000 | 4 | 2 | 0.25 | ✓ |
| 12 | dataset-12 | 1000 | 4 | 2 | 0.75 | ✓ |

### A.8.2 Results

We see similar trends on simulated datasets (Figure 16, 17, 18) compared to what we observed on UCI datasets.

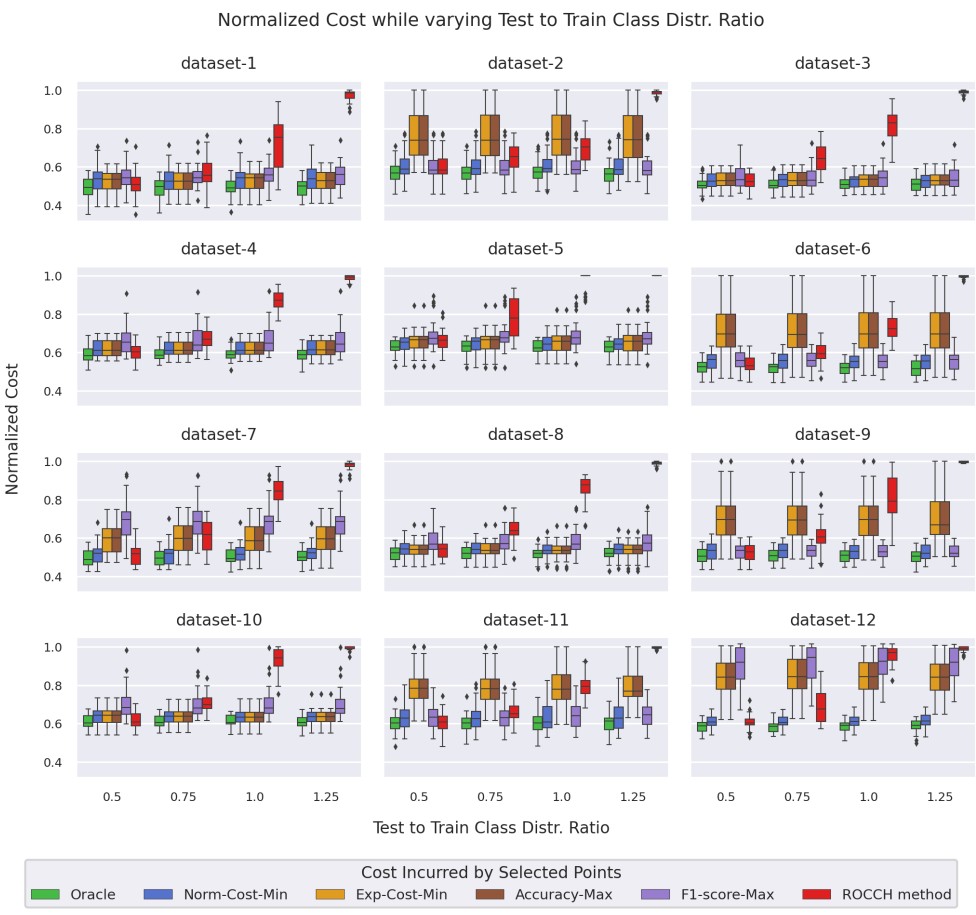

Figure 16: Effects of changing class distributions on model selection performance for 12 simulated datasets. Performance is quantified with expected cost. *Oracle* shows the cost of the best classifier; this is the target for the model selection methods.

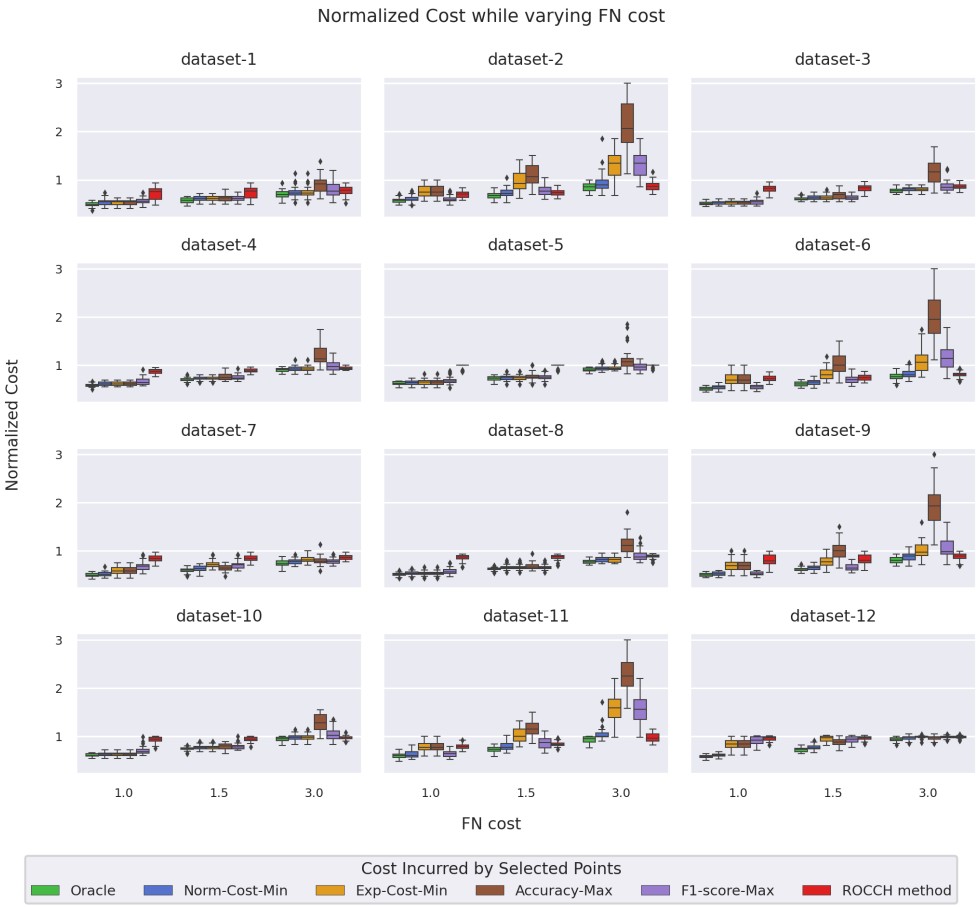

Figure 17: Effects of changing cost distributions on model selection performance for 12 simulated datasets. Performance is quantified with expected cost. *Oracle* shows the cost of the best classifier; this is the target for the model selection methods.

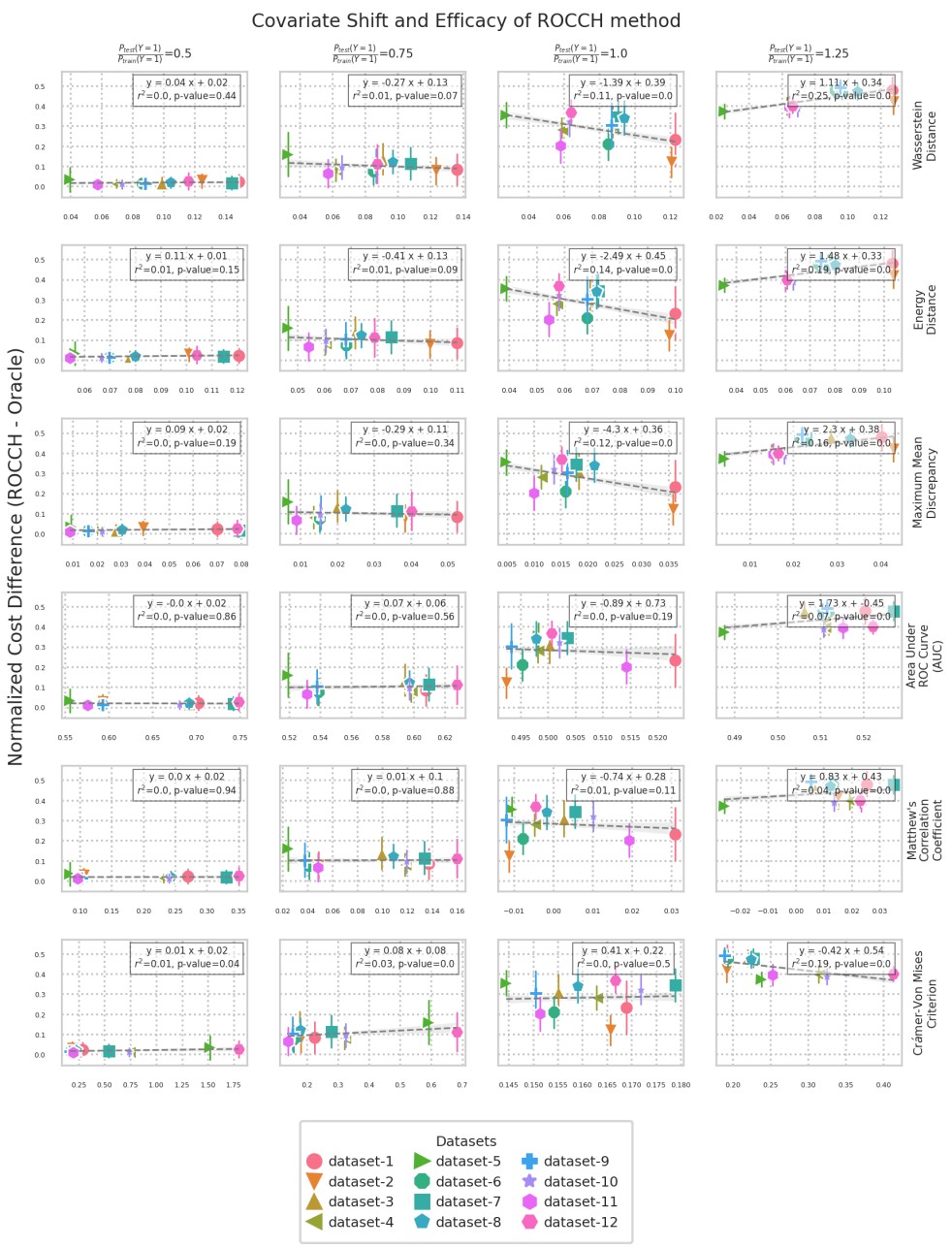

Figure 18: Effects of changing covariate shift on model selection performance, based on normalized cost, and applied to 12 simulated datasets. x-coordinates were derived by averaging covariate shift over all 30 runs.

