# OpenReview forum: "Model Selection of Discrete Classifiers under Class and Cost Distribution Change: An Empirical Study"
_TMLR — Rejected by TMLR_

### Review · Reviewer_orTE · 2023-08-12

**Summary Of Contributions:**

This paper considered model selection problem in the context of distribution shift, specifically the label distribution $P(Y)$ and cost distribution $C(\hat{Y}|Y)$. Concretely speaking, this model selection is based on the ROC curves. In the first step, ROC curves on different classifiers are drawn. Then a convex hull w.r.t. these classifiers will be constructed. Further an iso-performance line is obtained, then an optimal classifier could be obtained through the point of tangency of the corresponding classifier. Empirical evaluations are conducted through standard UCI datasets.


**Audience:**

Yes

**Claims And Evidence:**

No

**Requested Changes:**

I would strongly suggest a major revision on the paper. I really like the problem. However, I found that it is quite difficult and confusing to follow the paper. A clear and detailed math descriptions are strongly suggested.

**Strengths And Weaknesses:**

### Summary

In general, I really like the problem,  proposed idea and believe this paper addressed an interesting and important problem. However, due to the writing quality and clarity issue, I think this paper is NOT in a good shape for acceptance. I do think a major revision is necessary for the acceptance.

### Would some individuals in TMLR's audience be interested in the findings of this paper?

Yes. This paper considered the model selection issue under distribution shift. I do think it is important, interesting and worthy of publication.

### Are the claims made in the submission supported by accurate, convincing and clear evidence?

No. I do think a major revision is required. Below are some notable remarks/notes during the paper reading.


1. About paper’s structure/organization/notation

- What does it mean by Discrete Classifiers and continuous classifiers? Could you please provide a clear definition? In the classification task, we want a model to predict discret classes, thus every classifier should be a discrete classifier?

- Sec 2. This paper requires significant re-writing.
> First, we briefly review some basics of ROC analysis; Section A.1 provides a glossary of terms for less familiar readers. ROC space is described by false positive rate (FPR) on the x-axis and true positive rate (TPR) on the y-axis.

This is very weird. ALL the basic background information (related to your main contribution) and notation should be clearly defined in the main paper. Even it is well-known. Without a clear description, there are numerous concerns, what is x-axis, y-axis? Does this paper focus on binary or multi-class problems? What is $Y$, is there any relation to the y-axis. I would strongly suggest a rigor and clear revision.

> ROC curves are usually computed from predictions on validation data, from the same distribution as the training data. Discrete classifiers from different continuous classifiers lie on the same point in ROC space if they yield identical confusion matrices on the validation data.

Again. This should be clearly defined. For example, in general it is true validation data distribution = training data distribution. However, in **your scenario**, this does not necessarily hold. At this time, you should clearly define a validation, training, test set by a clear mathematical notation.

> In Figure 1, Step 2 shows the ROC convex hull of the three classifiers from Step 1. The QuickHull algorithm (Barber et al., 1993) can be used to generate the ROC convex hull of n classifiers in O(n log n) time. When the testing environment matches the training and validation conditions, one might select the hybrid classifier or one of the discrete classifiers it comprises, which span a range of trade-offs between FPR and TPR.

Again, without clearly presentation notations, it is a bit confusing to truly understand your point. What is the ROC convex hull? What is the definition/point by introducing this?

- In Figure 1, Step 4. I am not sure how the optimal classifier was decided. If I understand correctly (merely from figure 1(4)), we have the slope for the evaluated data, then draw the tangent line on the convex-hull curve? If the point of tangency lies in classifier 1, so we choose classifier 1, am I correct? Since there is no clear math, I have to guess what happened..

- Sec 3 has the similar issue of clarity. What does it mean by changed environment conditions/metrics?

2.  Technical concerns
- From figure 4, the model selection for test data should know the label distribution ratio in your proposed method? If so, it will be no surprise to have a better performance. Well, does this really fit the standard test dataset? Generally speaking, we should not have any information on test-dataset. Therefore, the selection should definitely consider worst-case scenarios in the validation dataset.

---

> ### Author Response · Authors · 2023-08-24
> **Thanks for your comments**
>
> We thank the reviewer for pointing out readability issues and we appreciate the enthusiasm for this research problem. The comments will help guide our revision to improve clarity.
>
> We understand that some topics may have been described too concisely in the main paper. In the revised paper, (i) we will introduce some mathematical notation to make things easier to follow, and (ii) move more background information from the appendix to the main paper.
>
> Detailed responses to comments are provided below.
>
> About paper’s structure/organization/notation:
>
> * Discrete classifiers refer to classifiers which can only predict discrete classes, whereas continuous classifiers refer to classifiers which predict a score (as mentioned in introduction and appendix).
> * In our case, the validation data is a subset of the training data therefore they have the same cost and class distributions. What does differ is the class and cost distribution of test environments.
> * The reviewer describes Step 4. correctly. To add a bit more description here, when drawing the tangent we iterate over each point on the hull to find the point which leads to the tangent (iso-performance line) with highest TPR (y-axis) intercept (with fixed slope we previously calculated).
> * When we say changed environment conditions we refer to test data such that the class distribution and associated cost distribution is different from the training/validation data.
>
> Technical concerns:
>
> Information for model selection is not supposed to come from the test dataset, rather it is meant to represent the expected conditions at test time. In our experimental settings we happened to use the exact class distribution calculated from the test data set. Authors of the ROCCH method formulated the problem setting of selecting classifiers for changed environments to be constrained by available expected information on test data. The other methods we investigated only access the same level of information on test data (class and cost distribution) as the ROCCH method, or less, but never more.
> Investigations on how methods perform when expected conditions do not match actual test conditions and their worst case scenarios are another interesting topic of research which we will mention in the Future Work section of our revised manuscript.

---

> > ### Comment · Reviewer_orTE · 2023-08-26
> > **About revised paper**
> >
> > Dear authors,
> >
> > I am wondering if you have updated the revised paper. If so, could you use different colors to highlight the revisions?
> >
> > Thank you

---

> > > ### Author Response · Authors · 2023-08-26
> > > **Revised paper**
> > >
> > > The revised paper has been uploaded. We have listed out the revisions we made along with where they are in the revised paper (Please kindly see Changes Since Last Submission).
> > >
> > > We can highlight the pdf if it is still needed.

---

> > > > ### Comment · Reviewer_orTE · 2023-09-08
> > > > **Follow-up question**
> > > >
> > > > Thanks. I still feel a bit confused about inherent assumptions within the paper. Would you clarify which distribution has been shifted and which distribution keep invariant. How this is related to ROC curve/F1 score? Which assumption is required if you can select the model during the test time?

---

> ### Author Response · Authors · 2023-09-11
> **Thanks for the follow-up questions**
>
> Thanks for the follow-up questions - we are eager to make sure these things are clear!
>
> >> Would you clarify which distribution has been shifted and which distribution keep invariant.
> * We anticipate that there will be a change/shift in test set w.r.t. the training set in terms of class distribution and cost distribution. Thus, $P_{train}(Y)$ may not be equal to $P_{test} (Y)$, and similarly $C_{train}(\hat{Y}|Y)$ may not be the same as $C_{test}(\hat{Y}|Y)$.
> We assume that the conditional distribution of observing labels Y given features X remains invariant across train and test sets, i.e, $P_{train}(Y|X) = P_{test}(Y|X)$. We have stated these assumptions in Section 2, specifically on the last paragraph of page 3.
> To summarize, we assume that the relationship between features and labels is preserved across train and test sets, but the rate at which we observe the labels and/or the relative costs of omission vs. commission mistakes might be different.
>
> >> How this is related to ROC curve/F1 score?
> * The ROC curve is composed of (FPR,TPR) points in ROC space. Provost and Fawcett (1997, 1998, 2001) introduced the ROC convex hull, which has the special property of being composed of the most “northwestern” available points in ROC space (these points may come from one or more ROC curves, or simply any collection of points). More northwestern points have lower FPR values (less false positive predictions) and higher TPR values (more true positive predictions). Thus, for a point on the ROC convex hull with a given FPR value, we are sure that the TPR value for that point is greater than the TPR value of any other available point with that given FPR value. The ROC convex hull method relies on isometrics of expected cost. Expected cost is calculated based on the class and cost distributions of the associated data (Equation 2 in Page 4). Isometrics assume: 1) that there are identifiable linear regions in ROC space where expected cost is equal; 2) that the slope of this isoperformance line can be calculated using the derivation in Section A.2; and 3) that the classifier that minimizes expected cost lies on the ROC convex hull where it is intersected tangentially by the isoperformance line. The ROCCH method utilizes these assumptions to select models for changed environments, while assuming that conditional distribution of observing Y given X will not change, whereas the rate of observing Y and cost of making mistakes on Y might be different. The F1-score is not directly related to these assumptions. We included it because of how ubiquitous it is in classifier evaluation. This is an experimental design choice rather than a choice motivated by or related to the assumptions of the ROCCH method.
>
> >> Which assumption is required if you can select the model during the test time?
> * The general assumption required for all model selection approaches we investigated is that $P_{train}(Y|X) = P_{test}(Y|X)$. The ROCCH method has its own specific assumptions for making model selections. The main assumption it makes is that the optimal classifier lies on the ROC convex hull where the isoperformance line that minimizes expected cost is tangential to the hull. We have discussed these assumptions in the beginning half of page 4 in the revised manuscript. The slope of isoperformance lines are calculated using the derivation in Section A.2 in the appendix. Our experimental results indicate that these assumptions made by the ROCCH method do not hold in most empirical settings, since the method underperforms relative to other approaches which do not rely on these assumptions.
>
> Do these explanations answer your questions? We are interested in anything that remains unclear, and especially in anything we could do differently in the sections noted in our answers above to make these points more clear. For example, is there particular notation that would help? Is there anything that you think needs to be moved from the appendix into the main paper? We are not sure whether we need to stay under the 12 page limit during revisions and for the camera-ready copy or if that is just a limit for the submission stage.
>
> References
>
> Provost, F., & Fawcett, T. (1997). Analysis and visualization of classifier performance with nonuniform class and cost distributions. In Proceedings of AAAI-97 Workshop on AI Approaches to Fraud Detection & Risk Management (pp. 57-63).
>
> Provost, F., & Fawcett, T. (1998, July). Robust classification systems for imprecise environments. In AAAI/IAAI (pp. 706-713).
>
> Provost, F., & Fawcett, T. (2001). Robust classification for imprecise environments. Machine learning, 42, 203-231.

---

### Review · Reviewer_mb9z · 2023-08-18

**Summary Of Contributions:**

The paper considers the task of finding a model that performs well under cost/label-shift when give a collection of (single/hybrid) models spanning the ROC convex hull. They compare some common methods to the ROOCH method that optimizes a the full expected cost using iso-performance lines.

**Audience:**

No

**Broader Impact Concerns:**

No broader impact concerns.

**Claims And Evidence:**

Yes

**Requested Changes:**

I mainly have questions that could instruct how to change the text / update experiments.1

1. Why aren't the accuracy or F1 considering the updated class-shift conditions? Is there a reason these metrics cannot use the known environment conditions the class probabilities? For example you can weight then accuracy in each class by the updated probability of the class and then average?

2. Why weren't larger datasets and more complicated models considered? What made the authors happy that they were not underfitting with their choice of models without hyperparameter tuning? The performance numbers on a bunch of tasks go as low as 0.2. Is that not cause for concern that the models are not learning as well as possible? Was something like boosting considered?

3. Why was cost only considered in the small (1 - 3) range? In domains like health care, a false negative can be considered cost 10 times as much as a false positive.

4. What was the largest change in class distribution between training and test?

Three and 4 seems important to ask because under large change one might not be able to construct good hybrid models. What guarantees good coverage of the convex hull, especially if I want use the paper's findings at scale and it is expensive to train many kinds of models.

5. Why is normalized cost the primary metric? It seems like the paper is saying "If you want to optimize this metric on the test data , you should use that metric to select models on the validation data". This sentence seems obvious to me. What subtlelty is there that I should take away from the results of the paper.

**Strengths And Weaknesses:**

1. The paper is well-written with useful plots and explanatory sentences.
2. The set of experiments in the paper covers a wide variety of tasks from the UCI ML repository.

---

> ### Author Response · Authors · 2023-08-24
> **Thanks for your comments**
>
> We thank the reviewer for their constructive comments. Please find the responses below.
>
> 1. When it comes to Accuracy and F1, we considered the common way of calculating them which does not include the weighting scheme you described.
> 2. We acknowledge that there are more complex models such as boosting methods which could produce better fitted classifiers. We focused more on how to select optimal models given that we have an existing set of trained models, rather than on producing the best models possible. This is why we used default hyperparameter values for the classifiers we used from the scikit-learn package. Hyperparameter search ranges would not be consistent across UCI data sets (due to different dataset sizes, no. of features, etc.), which would make comparing things more difficult.
> 3. We agree that there could be a wider range and finer granularity of experimental settings for consideration (as mentioned in the limitations). We could consider including experiments on wider cost ranges in the revised manuscript. We expect to see trends similar to what we are currently observing.
> 4. The largest change we simulated was when the test class distribution to train class distribution ratio was 0.5 (where the relative frequency of positives in test set was half of that of training set). The ROC convex hull is composed of north-western most points in ROC space. Therefore for any value of FPR, the point on the hull with x-coordinate FPR, will have the highest TPR compared to all other points with the same FPR. This ensures that all classifiers on the ROC convex hull are minimum cost classifiers by construction. We will add more discussion on theory around the ROC convex hull in the revised manuscript.
> 5. The authors of the ROCCH method suggest using normalized cost for empirical evaluation on test data. This is why we considered normalized cost as a primary metric. We will deemphasize the point about matching the optimization target and the metric in our revised manuscript. We should also add here that when we calculate the metric on which we base our model selection, we do so by first calculating class-wise performance on validation data (i.e., FPR and TPR on validation data) and then weigh these components by corresponding conditions of interest that will change in test data and we know beforehand, instead of conditions from validation data (which is the 'base' definition of the metric itself). The normalized cost we calculate on the validation set is a hybrid measure which incorporates both test data information (class and cost distribution of test data) and performance on validation data (FPR and TPR on validation data). The same analogy applies to when we calculated expected cost for model selection.

---

### Review · Reviewer_6CjN · 2023-08-20

**Summary Of Contributions:**

the paper studies two model selection techniques for the case where the train and test distributions are different.

Here, if I understand correctly, the scope is when Ptest(X|Y) = Ptrain(X|Y) but Ptest(Y) is different from Ptrain(Y). Model selection techniques may have access to Ptest(Y) at the time of model selection. In addition, in a cost-sensitive setting, the misclassification costs may change between train and test. Once again, the tests costs are available at model selection stage.

The authors implement two approaches: one based on ROC analysis and another one based on directly selecting the model that maximizes a specific metric on the validation set -- which may know test class distribution and costs. The authors provide experiments on UCI datasets with artificial distribution changes between train and test based on under/oversampling specific classes. They find that using directly optimizing on the validation set for the metic that will be used at test time works best.

**Audience:**

No

**Claims And Evidence:**

Yes

**Requested Changes:**

I think in order to be convincing, the authors should ground their discussion into a well-known task with a clear benchmark, where the common documented practice of model selection in this particular problem is ROCCH and they improve on the state-of-the-art with their change of model selection rule.

**Strengths And Weaknesses:**

Overall I do not believe the paper provides any insight that is worthy of publication. As far as I understand, the main findings:
-  "The most interesting and perhaps surprising take-away from our experiments is this: simply choosing the
classifier that optimizes performance on a validation set is often empirically better than model selection via
the ROCCH method, despite its theoretical grounding"
- "As noted above, the metric to be optimized on the validation set has substantial impact on the quality
of the model selection process."
do not really seem original or new to me. What the authors describe is the basic approach that practitioners are already doing (at least the ones I know) and ROCCH is just a graphical approach and interpretation for the specific case of cost-sensitive classification and its associated metric.

detailed comments
intro:
- the example of change in heart disease rate is a classic example of counterfactual estimation and many methods for that exist and have been empirically tested. Randomized experiments are usually necessary, and knowing the base rate of positive/negative examples in test data is likely insufficient in practice. Likely relevant to this discussion of changing environments are works involving causality (see Bottou et al. Counterfactual Reasoning and Learning Systems, JMLR 2012) and domain adaptation (see Zhang et al., Multi-Source Domain Adaptation: A Causal View, AAAI 2015).
- the discussion on precise/imprecise environment seems to be lacking critical assumptions here. For instance, do we assume P(x|y) to  be the same across train and test environments (Situation 3 in Zhang et al. 2015 above)? If not, knowing test class distributions and costs does not help much (at least as far as theoretical guarantees are concerned)
section 2:
- "ROC curves are usually computed from predictions on validation data, from the same distribution as the training data" -> Should we assume that this is the case considered by the authors when the  test distribution is different?
- "Given a set of discrete classifiers S that are on the ROC convex hull, our task is to select the classifier(s) s ∈ S that it optimizes some performance metric in a changed test environment." -> I would expect here some description of what is supposed to be *invariant* between train and test. If the test distribution can be arbitrary, the task is not really possible
- the normalised cost with C = 1 is also called "balanced error rate" if I understand correctly
section 3:
- I found Table 1 necessary to understand what metric used what information, I think it should be referenced at the beginning of the section
- I am missing what is nontrivial about algorithm 1 (model selection based on a specific metric on the validation set).
- why these metrics in particular? I'm also missing what is the goal here -- what is the relevant metric as far as the test task is concerned? Picking the best threshold for normalised costs only makes sense as far as we are interesting in normalised costs at test time (which we rarely are).
- why aren't all the metrics using the test class distribution? Since they can all be computed from the confusion matrix, they should all be able to incorporate this information
- is there a specific reason for the choice of F1 score? The cost corresponding to the optimal F1-classifier is usually dependent on the entire distribution and it's unclear why it's a good metric in the author's scenario  (see e.g. Parambath et al. Optimizing F-Measures by Cost-Sensitive Classification, NeurIPS 2014)
section 4.3
- " In such cases, we broke ties by selecting the discrete classifier which minimized cost among all discrete classifiers with the same (FPR, TPR)-coordinates." -> If the classifiers have the same values of both FPR and TPR, how can they have different costs? Or do you mean cost on test data? How would that be implemented in a model selection technique?
- I don't understand the different between exp-cost-min and ROCCH. I understand the procedure to select the classifiers is different, but I don't understand why they are not equivalent since both use  the same class distributions and costs
section 5:
- "the ROCCH method is sometimes competitive with Norm-Cost-Min." -> I don't understand the setting. Is the ROCCH calibrated to optimize for normalised or non-normalised costs using the test class distribution? My understanding from the previous sections was the latter. Then I don't understand why ROCCH it is a good or bad thing that ROCCH is competitive with Norm-Cost-Min if they do not have the same goals.
section 6:
- "The most interesting and perhaps surprising take-away from our experiments is this: simply choosing the
classifier that optimizes performance on a validation set is often empirically better than model selection via
the ROCCH method, despite its theoretical grounding" -> I am not sure this finding is worthy of publication. What the authors describe is the basic approach that practitioners are already doing (at least the ones I know) and ROCCH is just a graphical approach and interpretation for the specific case of cost-sensitive classification and its associated metric.
- "As noted above, the metric to be optimized on the validation set has substantial impact on the quality
of the model selection process." -> once again that is textbook knowledge and does not warrant a researchh publication.
- "In particular, we wondered whether our processes for generating changed test environments produced covariate shift" -> From the classification of Zhang et al. above, covariate shift is when P(Y|X) remains unchanged between environments, but P(x) changes. 2015 he approach of the authors to generate synthetic data follows P(X|Y) is the same across environments and thus should not follow the covariate shift assumption in general. If the authors have something else in mind with the phrase "covariate shift" I think making a proper definition would help.
section 7 related work:
- Even though the related work is enough for the scope of the paper, the generality of the introduction makes it necessary to be cite counterfactual approaches and domain adaptation

---

> ### Author Response · Authors · 2023-08-24
> **Thanks for your comments**
>
> We thank the reviewer for the thorough comments. The perspective and questions from this review have shown us some important ways to clarify our message and improve the paper.
>
> We agree that the first main finding listed is more interesting than the second. We will deemphasize the second bullet and focus more on why we think the first bullet is noteworthy. In short, the ROCCH method relies on the assumption that the minimum expected cost classifier lies on the iso-performance line corresponding to the computed slope. However, our empirical findings indicate that this assumption does not seem to hold in most settings, and that the minimum expected cost (and normalized cost) classifier lies somewhere else on the ROC convex hull. We further confirmed this when we measured the distance between classifiers selected by ROCCH method and Oracle in ROC space, and found them far away. We believe that this finding raises justified concern about reliance on isometrics in empirical settings.
>
> We believe we also need to clarify some details about the ROCCH method and the 4 other competing methods. Please note that the ROCCH method works differently than the other approaches.
> 1. The ROCCH method is not purely a visualization technique but actually a model selection technique that depends on theoretical cost equivalence and geometry to make selections, whereas the other methods were meant to resemble common practices in machine learning. However we do believe that there is some nuance to those practices which we tried to explore in our work, described below.
> 2. Because the ROCCH method relies on theoretical expected cost equivalence, we wanted to have a competing method which optimizes expected cost explicitly and empirically, hence Exp-Cost-Min. Please note that the ROCCH method and Exp-Cost-Min utilize the exact same test data information (as shown in Table 1), but in different ways, leading to different outcomes. It is also important to note that the ‘expected cost’ we measure here is a hybrid since it incorporates expected test conditions ($P(Y), C(\hat{Y}|Y)$ of test set) as well as performance on the validation data set (TPR and FPR on validation set).
> 3. The authors of the ROCCH method suggest using the normalized version of expected cost, i.e., normalized cost, to measure empirical performance on test sets, which led to us including Norm-Cost-Min in the list of competing methods. Similar to Exp-Cost-Min, this method is also a hybrid which incorporates expected test conditions and performance on validation data.
> 4. We included other model selection methods which optimize metrics that cannot explicitly incorporate class/cost distribution information into their calculation. Accuracy-Max was included since Accuracy is ubiquitous in classifier evaluation. We included F1-score-Max as another method which does not explicitly incorporate changed environment information and uses another commonly used metric for classifier evaluation.
>
> We believe that the subtlety of our finding lies in recognizing that the most empirically reliable way of making model selections for changed environments is to weight our validation set performance measure by the information on expected test conditions available to us (component-wise since FPR and TPR and individually weighted).
>
> We recognize that our results largely validate common practice. However, we find the failure of the ROCCH method noteworthy because the theory underlying the method would lead one to believe it to be a better choice. The ROCCH method does get some use across a variety of ML domains. For example, the method continues to find use in computational finance and multi-objective optimization. The significance of this result may vary across audience members depending on the extent to which they engage with the ROCCH, and in fact, this is part of the reason we submitted to TMLR, which “emphasizes technical correctness over subjective significance.” Since our search returned 250 citations using ROCCH in the past 3 years, we believe that some readers will be interested in the result.
>
> The reviewer had several comments referencing connections to counterfactual estimation and causal reasoning. We do see the connections and are happy to dedicate a paragraph in the Related Works section of the revised manuscript for discussing causality, domain adaptation, counterfactual estimation. However, we do see this setting as slightly different. The authors of the ROCCH method adopt a discriminative view while formulating the problem setting of changed environments, which we continue. Representation of causal subgroups and discussions on viewing changed environments through the lens of generative modeling have already been raised and addressed by the critics and authors of the ROCCH method (Webb and Ting, 2005; Fawcett and Flach, 2005). We have mentioned this back and forth correspondence in our Related Works section.

---

> > ### Author Response · Authors · 2023-08-24
> > **More details**
> >
> > Responses to some more specific comments are below:
> >
> > Introduction:
> >
> > We assume that conditional distribution of observing label Y given features X remains constant across train, validation, and test data sets, i.,e $P_{train}(Y|X) = P_{validation}(Y|X) =  P_{test}(Y|X)$.
> >
> > Section 2:
> >
> > The training and validation data always have the same distribution in our experiments, because common practice usually involves splitting the training data into a smaller sized training set and validation set. This holds true even when we simulate test class distributions to be different from training.
> >
> > Section 3:
> >
> > We will revise Section 3 to refer to Table 1 early on in the section.
> > Metrics on validation data only have access to confusion matrices on validation data and not to confusion matrices on test data. The label distribution is provided as an aggregated measure to applicable selection methods. This weighting by label distribution step is performed after TPR and FPR on the validation set are already computed.
> >
> > Section 4:
> >
> > The ROCCH method was originally designed to only select points on the ROC convex hull. We ended up with multiple classifiers in a single point because we chose to have multiple continuous classifiers to start with. To offset the effects of this experimental design choice, we broke these ties in the most advantageous way for the ROCCH method. In short, if the ROCCH method correctly located the FPR,TPR point on which the actual lowest test cost classifier lies, we considered it a correct model selection. This speaks to how we evaluated the model selection technique rather than the technique itself.
> >
> > Section 5:
> >
> > The ROCCH method has been shown to theoretically optimize non-normalized expected cost. However, we evaluate model selections using normalized cost, since the original authors suggested using this metric for empirical evaluation on test sets. In addition, we also investigated evaluating model selections using empirical expected cost (results are in the Appendix).
> >
> > Section 6:
> >
> > Our definition of covariate shift assumes that $P_{train}(Y|X) = P_{test}(Y|X)$ and that $P_{train}(X) \neq P_{test}(X)$. We will add this information to the revised manuscript.
> >
> > References
> >
> > Webb, G. I., & Ting, K. M. (2005). On the application of ROC analysis to predict classification performance under varying class distributions. Machine learning, 58, 25-32.
> >
> > Fawcett, T., & Flach, P. A. (2005). A response to Webb and Ting’s on the application of ROC analysis to predict classification performance under varying class distributions. Machine Learning, 58, 33-38.)

---

### Decision · Action_Editor_QEZH · 2023-10-29

**Recommendation:** Reject

**Comment:**

As discussed earlier, the main concerns on the paper are - (i) the method proposed as a better alternative is rather straightforward and already a common practice, (ii) experiments on UCI datasets with synthetically injected shifts aren't sufficiently convincing and do not provide insights into when this method might fail to work.

**Audience:**

Researchers working on cost-sensitive optimization or model selection under distribution shifts will be interested in the paper.

**Claims And Evidence:**

The paper considers the problem of model selection under distribution or environment changes where p(y|x) is assumed to be the same across train and test environments. It evaluates two methods for this problem: an existing ROC convex hull (ROCCH) method, and a method that directly uses a validation metric adjusted using the test domain costs and/or priors.

The paper makes two main claims. The first claim is that ROCCH method which is specifically designed for this problem does not perform well empirically in the authors' experiments. The second claim is that simply using a validation set metric adjusted with test priors and costs performs better than ROCCH. Authors show experiments on UCI datasets with synthetic shifts to support these claims.

A major concern from reviewer 6VjN is that the method proposed as a better alternative for the problem is straightforward and is already a common practice, and thus may be of low interest to TMLR's audience. The authors mention that ROCCH still finds use in computational finance and multiobjective optimization but the experiments in the paper do not consider real datasets from these domains.

There were some other concerns from the reviewers about lack of clarity in assumptions the paper makes which have since been addressed in the authors' response. An outstanding concern is lack of characterization/insights on when the method is expected to work. The paper experiments with mild shifts in the class priors but experimentally testing the method on larger shifts might help in getting more insights into why/when the method is expected to work.